# Advanced Pathogen Monitoring in *Penaeus vannamei* from Three Latin American Regions: Passive Surveillance Part 2

**DOI:** 10.3390/v17020187

**Published:** 2025-01-28

**Authors:** Pablo Intriago, Bolivar Montiel, Mauricio Valarezo, Jennifer Gallardo, Yamilis Cataño

**Affiliations:** 1South Florida Farming Corp., 13811 Old Sheridan St, Southwest Ranches, FL 33330, USA; 2South Florida Farming Lab., Av. Miguel Yunez, km 14.5 via a Samborondón, Almax 3 Etapa 1 Lote 3 Bodega 2, Samborondón CP 092302, Guayas, Ecuador; bolivarmontielr@gmail.com (B.M.); mauriciovalarezogilbert@gmail.com (M.V.); jenniferg_nc@hotmail.com (J.G.); 3Océanos S.A., Centro de Producción Laboratorio, Coveñas, Colombia; silimaya@yahoo.es

**Keywords:** Wenzhou shrimp virus 8—WzSV8, hepanhamaparvovirus—DHPV, infectious hypodermal and haematopoietic necrosis virus—IHHNV, *Enterocytozoon hepatopenaei*—EHP, necrotizing hepatopancreatitis bacteria—NHPB, white spot syndrome virus—WSSV, postlarvae—PL, viral inclusion—VIN

## Abstract

This study presents the second phase of a year-long investigation comparing multiple PCR analyses and histological examinations to confirm the presence of characteristic lesions of each pathogen in three different regions of Latin America. More than 20 agents, including DNA and RNA viruses, bacteria and microsporidia, have been targeted. In addition to wild *Penaeus vannamei*, which was studied previously, samples of wild *P. stylirostris* and *P. monodon* were included. Notably, a positive PCR test result alone does not confirm the presence of a viable pathogen or a disease state. Similarly, positive PCR results do not necessarily correlate with the presence of histological lesions characteristic of the targeted pathogen. Wenzhou shrimp virus 8 (WzSV8) was found to be widespread among shrimp in all regions, including both farm-raised and wild populations. Histopathological analysis indicated that shrimp typically presented coinfections, such as WzSV8, Decapod hepanhamaparvovirus (DHPV), chronic midgut inflammation, and tubule distension/epithelial atrophy, consistent with the toxicity of Pir A/B or another bacterial toxin. Bacterial muscle necrosis was also found in some regions. In general, bacterial infection was the dominant pathology in all three regions during the year. We also postulate that both WzSV8 and DHPV can infect not only hepatopancreatic cells but also cells in the ceca and intestine.

## 1. Introduction

This study constitutes the second phase of a year-long investigation focused on comparing multiple PCR analyses and histological examinations to confirm the presence of characteristic lesions associated with shrimp pathogens across three distinct regions of Latin America, as described by [1]. Building upon the findings from the initial phase, this comprehensive evaluation targeted over 20 pathogens, encompassing DNA and RNA viruses, bacteria, and microsporidia.

The DNA viruses assessed in this study include *Hepatopancreatic parvovirus* (HPV), renamed *Decapod hepanhamaparvovirus* (DHPV) [2]; *Macrobrachium hepatopancreatic bidnavirus* (MHBV) [3]; *Decapod iridescent virus 1* (DIV1) [4]; white spot syndrome virus (WSSV); and *Infectious hypodermal and haematopoietic necrosis virus* (IHHNV), renamed *Penstylhamaparvovirus 1* [2], but referred to herein as IHHNV.

The RNA viruses examined include *Wenzhou shrimp virus 8* (WzSV8) [5], later renamed *Penaeus vannamei Picornavirus* (PvPV) [6], and subsequently *Penaeus vannamei solinvivirus* (PvSV) [7], but referred to herein as WzSV8; *Penaeus vannamei nodavirus* (PvNV) [8]; *Covert mortality nodavirus* (CMNV) [9]; *Infectious myonecrosis virus* (IMNV) [10,11]; *Yellow head virus* (YHV) [12]; *Taura syndrome virus* (TSV) [13]; and *Macrobrachium rosenbergii nodavirus* (MrNV) [14], which is also known to infect *Penaeus vannamei* [15].

The bacterial pathogens and microsporidia studied include intracellular bacteria such as *Spiroplasma* [16], rickettsia-like bacteria (RLB) [17,18], *Necrotizing hepatopancreatitis bacteria* (NHPB) [19], and extracellular bacteria, including *Vibrio* species [20]. The microsporidium *Enterocytozoon hepatopenaei* (EHP) [21] and other non-EHP microsporidia were also analyzed [22,23].

### 1.1. Findings from Phase One

The initial phase of this study underscored the critical importance of primer selection for obtaining reliable diagnostic results [1]. Significant variations were observed among different primer sets, emphasizing the need for seasonal evaluations to identify the most effective combinations. Inconsistent outcomes with several primers for viral pathogens were attributed to factors such as pathogen genetic diversity, geographic variations in host populations, and environmental influences. Moreover, false-positive PCR results were potentially linked to endogenous viral elements (EVE) within the shrimp genome, which could harbor target sequences.

*Wenzhou shrimp virus 8* (WzSV8) emerged as the most frequently detected pathogen across all sampled regions, regardless of shrimp size or environment, including wild populations. Co-infections with bacterial lesions and other pathogens such as DHPV, WSSV, or BP were common, particularly in midgut tissue. Interestingly, WzSV8 was also detected in wild shrimp populations (Region 1) at a 100% prevalence rate, suggesting its endemic nature in this region and its widespread presence globally.

Notably, differences in co-infection rates were observed between the three regions, influenced by culture systems and the health status of broodstock or postlarvae sources. Midgut histopathology results were consistent with conditions such as SHPN, chronic AHPND, and/or RLB/NHP, aligning with previous reports that *Vibrio*-associated diseases predominate in the region [24,25].

### 1.2. Sample Context and Rationale

Most samples were submitted by clients for health monitoring or disease outbreak investigations, and while broodstock specimens generally appeared healthy, they served as critical reference points for evaluating pathogen transmission through post-larvae. Additionally, wild penaeid shrimp (*P. vannamei*, *P. stylirostris*, and *P. monodon*) were included to explore their potential roles in pathogen reservoirs and interactions with aquaculture.

This study is not an epidemiological investigation but rather a prevalence analysis of shrimp pathogens in randomly collected samples from hatcheries, farms, maturation units, and wild populations across three Latin American regions between May and October 2023. It represents the second phase of a one-year project, expanding on findings from the initial report [1], with a focus on histology and PCR for identifying pathological variations.

## 2. Materials and Methods

### 2.1. Sample Collection

More than 140 samples from surveillance sampling of *P. vannamei* originating from three different regions in Latin America were analyzed throughout the period spanning from May 2023 to October 2023. One region draws its culture water from the Atlantic Ocean, and the other two from the Pacific Ocean. Regions 1 and 3 were the same regions as described in the first surveillance described in [1], whereas the third region was a different region; for this reason, we named it Region 4. Regions 1 and 4 were in the Pacific Ocean, and Region 3 was in the Atlantic Ocean.

Sampling, when possible, included animals from hatcheries, broodstock centers, farms, and wild animals. It should be noted that shrimp sampled for PCR and histology were different individuals from the same populations. To protect client privacy, the countries or exact locations from which the samples were obtained are not revealed here. However, the clients from World Organization for Animal Health (WOAH) member countries were informed of their responsibility to notify the competent authority of their country regarding positive test results for any shrimp pathogens listed by the WOAH or arising from any unusual incidences of mortality. It would then be the responsibility of the relevant competent authorities from those member countries to report to the WOAH.

#### Sampling Locations and Data Collection

Samples were collected over a period of six months from three regions: Region 1 and Region 4 on the Pacific Coast and Region 3 in the Caribbean Sea (Appendix A).
Region 1: Samples were taken from four different hatcheries (5 samples in total), eight company farms (76 samples), six broodstock rearing units (44 samples), and from live wild shrimp obtained from local fishermen, including *Penaeus vannamei* (10 samples) and *Penaeus stylirostris* (8 samples).Region 3: Samples were collected from the country’s only hatchery (3 samples) and the only farm (10 samples). Additionally, live wild *Penaeus monodon* (10 samples) were purchased from local fishermen.Region 4: Samples were collected from two hatcheries (5 samples in total), two farms (4 samples), and one maturation unit.

It is important to note that not all the samples were divided and processed for both histology and PCR. Some samples were submitted for either histology or PCR, while others were used for both analyses, depending on the source and submission conditions. Farmers typically sent samples without specifying whether they were associated with disease outbreaks or any indication of a problem, and no samples were explicitly labeled as diseased. As a result, the study was conducted without prior knowledge of the health status of the shrimp, relying solely on the results of laboratory analyses.

### 2.2. PCR Methods Used

DNA was extracted from whole larvae, tissue, or organs fixed in 90% alcohol, following the manufacturer’s protocol (Omega, Bio-Tek E.Z.N.A. tissue DNA kit; Omega, Bio-Tek Inc., Norcross, GA, USA). In brief, each sample was minced with sterilized scissors and then ground using a microcentrifuge pestle. Approximately 200 mg of tissue was then transferred to a clean 1.5 mL Eppendorf tube. To this end, 500 μL of tissue lysis buffer (TL) and 25 μL of Omega Biotek (OB) protease solution were added, and the mixture was vortexed and then incubated in a thermoblock at 55 °C for approximately 3 h, with vortexing every 30 min. RNA was removed by adding 4 μL of RNase A (100 mg/mL), and after mixing, the sample was kept at room temperature for 2 min. The sample was then centrifuged at 13,500 RPM for 5 min, and the supernatant was carefully transferred to a new 1.5 mL Eppendorf tube. Then, 220 μL of BL buffer was added, and the mixture was vortexed and incubated at 70 °C for 10 min. Next, 220 μL of 100% ethanol was added, the mixture was vortexed, and the contents were passed through a HiBind^®^ DNA Mini Column into a 2 mL collection tube. The columns were then centrifuged at 13,500 RPM for 1 min, after which the filtrate was discarded. Subsequently, 500 μL of HBC buffer (diluted with 100% isopropanol) was added to the column, and the sample was spun at 13,500 RPM for 30 s. The filtrate was discarded, the column was washed twice with 700 μL of DNA wash buffer diluted with 100% ethanol, and the sample was centrifuged at 13,500 RPM for 30 s. The filtrate was discarded. This step was repeated. The column was then centrifuged at 13,500 RPM for 2 min to dry it. The dried column was placed in a new nuclease-free 1.5 mL Eppendorf tube, and 100 μL of elution buffer, which was heated to 70 °C, was added to the column. The sample was allowed to sit for 2 min before being centrifuged at 13,500 RPM for 1 min. This elution step was repeated. The eluted DNA was then stored at −20 °C until needed.

RNA was extracted from whole larvae, tissue, or organs fixed in 90% alcohol, following the manufacturer’s protocol (Omega, Bio-Tek E.Z.N.A. Total RNA Kit). In brief, each sample was minced with sterilized scissors and then ground using a microcentrifuge pestle. Approximately 200 mg of tissue was then transferred to a clean 1.5 mL Eppendorf tube. Then, 700 μL of TRK Lysis Buffer was added, and the tube was left at room temperature for approximately 3 h, with vortexing every 30 min. The sample was then centrifuged at 13,500 RPM for 5 min, and the supernatant was carefully transferred to a new 1.5 mL Eppendorf tube, to which 420 μL of 70% ethanol was added. After vortexing to mix thoroughly, the contents were passed through a HiBind^®^ RNA Mini Column into a 2 mL collection tube. The columns were then centrifuged at 13,500 RPM for 1 min, after which the filtrate was discarded. Subsequently, 500 μL of RNA Wash Buffer I was added to the column, and the sample was spun at 13,500 RPM for 30 s. The filtrate was discarded, and the column was washed twice with 500 μL of RNA Wash Buffer II and diluted with 100% ethanol. The column was then centrifuged at 13,500 RPM for 1 min to dry it. The filtrate was discarded. This step was repeated. The column was then centrifuged at 13,500 RPM for 2 min to dry it. The dried column was placed in a new nuclease-free 1.5 mL Eppendorf tube, and 70 μL of nuclease-free water was added to the column. The sample was centrifuged at 13,500 RPM for 2 min. This elution step was repeated. The eluted RNA was then stored at −70 °C until needed. The samples used for extraction were as follows:
Samples used for extraction
IHHNV: 2 pleopods per animal pool of 5 animals.PvNV: 2 pleopods per animal pool of 5 animals.Spiroplasma: DNA pool of 2 pleopods per animal, pool of 5 animals; 10 gill pools of animals; whole hepatopancreas pools of 5 animals.WSSV: 10 gills per animal pool of 5 animals.TSV: 10 gills per animal pool of 5 animals.MrNV and XSV: 10 gills per animal pool of 5 animals.IMNV: 10 gills per animal pool of 5 animals.YHV-GAV: 10 gills per animal pool of 5 animals.DHPV: Whole hepatopancreas pool of 5 animals.MHBV: Whole hepatopancreas pool of 5 animals.DIV1: Whole hepatopancreas pool of 5 animals.WzSV8: Whole hepatopancreas pool of 5 animals.PvSV: Whole hepatopancreas pool of 5 animals.RLB: Whole hepatopancreas pool of 5 animals.NHPB: Whole hepatopancreas pool of 5 animals.EHP: Whole hepatopancreas pool of 5 animals.

AHPND: Whole hepatopancreas pools of 5 animals. Microsporidia: 0.5 g of tail muscle per animal pool of 5 animals.

CMNV: 0.5 g of tail muscle per animal pool of 5 animals.

Vibrio Community: DNA pool of 2 pleopods per animal pool of 5 animals; 0.5 g of tail muscle per animal pool 5 of animals.

### 2.3. The Following Pathogens Were Screened with the Methods Tested

Wenzhou shrimp virus 8 (WzSV8) was identified using [7,26,27]. Hepanhamaparvovirus (DHPV) was identified following Phromjai et al. [28,29,30,31]. Macrobrachium Bidnavirus (MrBdv) followed [32], and rickettsia-like bacteria (RLB) [17,18,33]. Necrotizing hepatopancreatitis bacteria (NHP-B) followed [19]. *Spiroplasma* followed [16,34,35]. Non-EHP Microsporida was identified with [21,22,23,36,37,38]. Infectious hypodermal and hematopoietic necrosis virus (IHHNV) were identified using [39,40,41,42]. *Enterocytozoon hepatopenaei* (EHP) was identified following [21,42,43,44,45]. *Vibrio* spp. was identified following [46]. Acute hepatopancreatic necrosis (AHPND) was identified following [47]. Decapod iridescent virus 1 (DIV1) was identified following [4]. White spot syndrome virus (WSSV) was identified following [48]. Penaeus vannamei Noda virus (PvNV) was identified following [49]. Covert mortality Nodavirus (CMNV) was identified following [9]. Infectious myonecrosis virus (IMNV) was identified following [8]. Yellow head virus (YHV) was identified following [50]. Taura syndrome virus (TSV) was identified following [13,51]. Machrobrachium Nodavirus (MrNV) was identified applying [3]. Extra small virus XSV was identified using [15].

### 2.4. Primer Sets Used in This Study

The primers used for PCR analysis were synthesized and manufactured by Life Technologies, ThermoFisher Scientific (Waltham, MA, USA).


**Primer**

**Product**

**Sequence (5′-3′)**

**Ta**

**References**

**WSSV**




146F1146R1.146F2 Nested146R2 Nested1447 bp941 bpACTACTAACTTCAGCCTATCTAGTAATGCGGGTGTAATGTTCTTACGAGTAACTGCCCCTTCCATCTCCA.TACGGCAGCTGCTGCACCTTGT55 °C55 °C[48]
**DHPV**




H441F1H441R1441 bpGCATTACAAGAGCCAAGCAGACACTCAGCCTCTACCTTGT60 °C[28,29]HPVnFHPVnR1265 bpATAGAACGCATAGAAAACGCTCAGCGATTCATTCCAGCGCCACC55 °C
HPVnFHPVnR265 bpATAGAACGCATAGAAAACGCTGGTGGCGCTGGAATGAATCGCTA55 °C
1120F1120R592 bpGGTGATGTGGAGGAGAGAGTAACTATCGCCGCCAAC60 °C[30]DHPV-U 1538 FDHPV-U 1887 RDHPV-U 1622 F.350 bp266 bpCCTCTTGTTACATTTTACTCGATGTCTTCTGTAGTCCAAGTTTGCACAGTGGTTGT55 °C55 °C[31]
**IHHNV**




389F389 bpCGGAACACAACCCGACTTTA55 °C[42]389R
GGCCAAGACCAAAATACGAA

77012F356 bpATCGGTGCACTACTCGGA55 °C[39]77353R
TCGTACTGGCTGTTCATC

392F392 bpGGGCGAACCAGAATCACTTA55 °C[41]392R
ATCCGGAGGAATCTGATGTG

309F309 bpTCCAACACTTAGTCAAAACCAA55 °C[42]309R
TGTCTGCTACGATGATTATCCA


**MrBidnavirus**




MrBdv-LMrBdv-R392 bpGCATTAATGGATTGGGAAGGTCGATGTCTGGATGACCGTA53 °C[32]
**DIV1**




SHIV-F1SHIV-R1SHIV-F2SHIV-R2457 bp129 bpGGGCGGGAGATGGTGTTAGATTCGTTTCGGTACGAAGATGTACGGGAAACGATTCGTATTGGGTTGCTTGATCGGCATCCTTGA59 °C59 °C[4]
**PvNV**




PvNV339FPvNV339RPvNV246NFPvNV246NR339 bp246 bpCTGTCTCACAGGCTGGTTCACCGTTTGAATTTCAGCAACACAAAACTGTGCCTTTGATCGGCCTTATCCACACGAACGTC55 °C60 °C[49]
**IMNV**




4587F4914R4725NF4863NR328 bp139 bpCGACGCTGCTAACCATACAAACTCGGCTGTTCGATCAAGTGGCACATGCTCAGAGACAAGCGCTGAGTCCAGTCTTG60 °C65 °C[8]
**CMNV**




CMNV-7F1CMNV-7R1CMNV-7F2CMNV-7R2619 bp165 bpAAATACGGCGATGACGACGAAGTGCCCACAGACCACAACCGAGTCAAACCGCGTAAACAGCGAAGG45 °C50 °C[9]
**TSV**




9992 F9195 R7171 F7511 R231 bp341 bpAAGTAGACAGCCGCGCTTTCAATGAGAGCTTGGTCCCGACAGTTGGACATCTAGTGGAGCTTCAGACTGCAACTTC60 °C60 °C[13,51]
**YHV-GAV**




YHV GY1YHV GY4YHV GY2YHV Y3YHV G6YHV GY5794 bp406–277 bpGACATCACTCCAGACAACATCTGGTGAAGTCCATGTGTGTGAGACGCATCTGTCCAGAAGGCGTCTATGAACGCTCTGTGACAAGCATGAAGTTGTAGTAGAGACGAGTGACACCTATGAGCTGGAATTCAGTGAGAGAACA66 °C66 °C[50]
**WzSV 8**




504F504R170F (Nested)170R (Nested)428 F BIOT428 R BIOT168 F BIOT Nested168 R BIOT Neste504 bp170 bp482 bp168 bpCAAGGTGGAGGTTACAGGAGACGAGGTATCCGTTGATGTCGACCGATGAATACGACAGAGAGGACAAGAGGAAGATTTGGCATGCCTCTGGAAAGCGATACGGTGTTAGATCGCTCCTTCTTCGAAAGCGATACTCCTACGACAGTCTTGAGTTTGAGGAAGGTGAG60 °C60 °C60 °C60 °C[26][27]
**PvSV**




3136 F3268 R133 bpTACGCCACGAACGAGAACAAGGACAGCGACAAAGACGAGA60 °C[7]
**MrNV**




fragment1Ffragment1Rfragment2-Ffragment2-RRNA2-fragm1FRNA2-fragm1RRNA2-fragm2-FRNA2-fragm2-RFL-XSV-FFL-XSV-RMrNV-RNA2 FMrNV-RNA2 RXSV FXSV R1486 bp1736 bp664 bp534 bp681 bp796 bp507 bpGTTAAACGTTTTGTTTTCTAGCACACCTACATTCGCTTCGGGCCCGAAGCGAATGTAGGTGTCGAAAGAGTGAAGGAGACTTGGCCCATCATGTGCTAGATATGACAGGCAGGCTACGTCACAAGTACTTGTGACGTAGCCTGCCTAAAGGATATTCGATATTCTATCCCACGTCTAGCTGCTGAC GTTAAGGTCTTTATTTATCGACGCGATACAGATCCACTAGATGACCGACGATAGCTCTGATAATCCGGAGAACCATGAGATCACGCTGCTCATTACTGTTCGGAGTC50 °C50 °C50 °C50 °C50 °C50 °C52 °C[3][15]
**EHP**




EHP510 FEHP510 RMF1MR1VE-SWP-365FVE-SWP-365Rβ-tubulin-262Fβ-tubulin-879RSWP_1FSWP_1RSWP_2F NestedSWP_2R NestedSSU ENF779 F1SSU ENF779 R1SSU ENF176 F1SSU ENF176 R1510 bp900–1000 bp365 bp262 bp514 bp148 bp779 bp176 bpGCCTGAGAGATGGCTCCCACGTGCGTACTATCCCCAGAGCCCGACCG GAG AGG GAG CCT GAGAGAC GGG CGG TGT GTA CAAATTCATGCAGATACAGCATTTGTAATTACGCCATTTATCATGCTTCAGCTGGTTGAAAATGCAAAGTGCAAAAATGCCTTTCGTTTTGCAGAGTGTTGTTAAGGGTTTCACGATGTGTCTTTGCAATTTTCTTGGCGGCACAATTCTCAAACAGCTGTTTGTCTCCAACTGTATTTGACAGCAGGCGCGAAAATTGTCCAAAGAGATATTGTATTGCGCTTGCTGCAACGCGGGAAAACTTACCAACCTGTTATTGCCTTCTCCCTCC60 °C55 °C60 °C60 °C58 °C64 °C58 °C64 °C[43][21][44][44][45][42]
**Microsporidia**




TS1TS218f1492rV11492600 bp1200 bp1200 bpGTCGGAATTCGCCAGCAGCCGCGGTCAGCGGATCCGTCAAATTAAGCCGCCACCAGGTTGATTCTGCCTGACGGTTACCTTGTTACGACTTCACCAGGTTGATTCTGCCGGTTACCTTGTTACGACTT55 °C45 °C58 °C[22,23][36,37][38]
**Rickettsia**




BACT FBACT RRIK FRIK RRp877pRp1258 n1500 bp532 bp380 bpCCGAATTCGTCGACAACAGAGTTTGATCCTGGCTCAGCCCGGGATCCAAGCTTACGGCTACCTTGTTACGACTTGCGTAGGCGGATTAGTTAGTCAGAGGTTGCGCTCGTTACAGGACTGGGGACCTGCTCACGGCGGATTGCAAAAAGTACAGTGAACA45 °C60 °C45 °C[13][33]
**NHPB**




NHPF2NHPR2379 bpCGTTGGAGGTTCGTCCTTCAGTGCCATGAGGACCTGACATCATC72 °C[19]
**Spiroplasma**




CSF:5′CSR:5′F28R5Pri-1Pri-2269 bp270 bp1200 bpTAGCCGAACTGAGAGGTTGAGATAACGCTTGCCACCTATGCGCAGACGGTTTAGCAAGTTTGGGAGCACCGAACTTAGTCCGACACTTGCTGATTCGCGATTACTAGCTAATACATGCAAGTCGAACG60 °C56 °C65 °C[16][34][35]
**Vibrio**
120 bp


Vib-FVib2-R
GGCGTAAAGCGCATGCAGGTGAAATTCTACCCCCCTCTACAG55 °C[46]
**AHPND**
1269 bp


AP4F1AP4R1AP4F2 NestedAP4R2 Nested230–357 bp1142–1269 bpATGAGTAACAATATAAAACATGAAACACGATTTCGACGTTCCCCAATTGAGAATACGGGACGTGGGGTTAGTCATGTGAGCACCTTC55 °C55 °C[47]

### 2.5. Histopathology

For histological analysis, samples were prepared following the procedures outlined by [52]. Briefly, the samples were fixed in Davidson’s AFA for at least 24 h, or 72 h in the case of broodstock, before processing for routine histological analysis of 5 µm thick tissue sections stained with hematoxylin and eosin (H&E). In addition, methyl green-pyronin modified stain was also used to distinguish DNA and RNA. From each juvenile to broodstock shrimp sample, 2 to 4 paraffin blocks were prepared. For postlarvae (PL), approximately 1500 animals were taken from each tank, and more than half (i.e., approximately 750+) were fixed in Davidson’s AFA for 24 h and embedded in paraffin blocks, each containing approximately 100 to 250 PL.

## 3. Results

### 3.1. PCR Results

In Region 1, as in the other two regions, WzSv8 was the most important pathogen detected. Like previous studies, both the UAZ and Biotech primers [7,27] produced the highest prevalence (Table 1). In Region 4, WzSV8 was present in 100% of the samples tested. The average prevalence of WzSV8 in all three regions was 82%. WzSv8 was detected in the PL, farm animal, and broodstock of *P. vannamei*. It was found in wild *P. vannamei*, *P. stylirostris*, and *P. monodon*.

The hepanhamaparvovirus (DHPV) was most common in Regions 1 and 4. Region 3 was detected in only one sample of PL. DHPV was most found in farm animals, followed by broodstock. In contrast to WzSV8, DHPV was never detected in wild stocks (Table 2). In the first study [1], we included Macrobrachium Bidnavirus (MrBdv) because of its histological similarity to HDPV lesions. However, it was never detected. The most effective primers for DHPV were those reported by [28,29].

The most common PCR screening tools for intracellular bacteria in shrimp are *Rickettsia*-like organisms (RLOs) [17] and the necrotizing hepatopancreatitis bacterium NHPB [19] (Table 3). We compared these two methods with the *Rickettsia* genus-specific primers Rp877p/Rp1258n [33]. In general, only RLO was detected, and the *Rickettsia* genus-specific primers *Rp877p/Rp1258n* were most prevalent.

We three tested methods to detect Spiroplasma followed the studies [16,33,34]. The highest prevalence was obtained using *S. penaei* primers [16], averaging 20% prevalence in the three regions, followed by *S. mirum* primers [35]. The average prevalence was 9% in the three regions. The prevalence of *Spiroplasma* ribosomal DNA in Creutzfeldt–Jakob disease (CJD) was low when primers were used [34], and Spiroplasma ribosomal DNA was present only in Region 1. In general, this pathogen was detected mainly in farm animals, with a very low prevalence (1/15) in PL and two cases in broodstock (2/37) (Table 4). No positive bacteria were detected in the wild animals. There was no detection in Region 4 at any stage.

The presence of non-EHP microsporidia (Table 5) was determined using three methods. The first method, developed by [22], was designed to detect *Agmasoma penaei* in *P. merguensis* and *P. monodon*. This method yielded the highest prevalence, averaging 85% across all three regions. Although the lowest concentration was observed in PL, positive results were obtained for all stages, including farm animal, broodstock, and wild samples. We also tested a more recent method [36] designed to be selective for *A. penaei* in *Litopenaeus setiferus* from the Gulf of Mexico. However, this method showed the lowest prevalence, averaging 21% across the three regions. Among farm animals, PL, broodstock, and wild animals had very low or nonexistent prevalence, while the highest prevalence (80%) was found in Region 4. The third method utilized universal primers widely used for microsporidians [38] and provided intermediate results, with an average prevalence of 47% across the three regions. The lowest prevalence was observed in the PL samples.

When testing IHHNV (Table 6), it is necessary to distinguish viruses from endogenous viral elements (eve). For this reason, we used all four PCR primers [39,40,41,53]. All these viruses must be present for a functional virus to replicate. However, if some but not all primers amplify, this pattern is a clear indication that the PCR primers are measuring an eve and not the complete virus.

The primers with the highest and lowest prevalence in descending order were 309 F/R, 392 F/R, 389 F/R, and 77012 F/77353R, averaging 45%, 40%, 39%, and 24%, respectively, in the three regions. These results revealed average virus and eve prevalence rates of 23% and 26%, respectively, in the three regions. Interestingly, *P. monodon* from Region 3 showed zero prevalence of IHHNV and 50% prevalence of eve, but when primers for an exclusive sequence for eve in monodon MG831F/R primers were used [41], all animal samples were positive. Except for one sample (1/3) of PL in Region 3, all farm animals and broodstock were negative.

Based on a previous study by Intriago et al. [1], we tested different sets of primers for the detection of Enterocytozoon hepatopenaei (EHP). Table 7 summarizes the results obtained from six different primers used in our tests. Generally, most existing PCR detection methods target the EHP small subunit ribosomal RNA (SSU rRNA) gene (SSU-PCR). However, our results clearly indicate that these methods often produce false positives due to the cross-reactivity of SSU-PCR primers with DNA from closely related microsporidia. For now, we consider the method targeting the spore wall protein (SWP) gene of EHP, as described by [45], to be the most reliable. This method minimizes false positives and provides a more accurate detection of EHP.

The white spot syndrome virus (WSSV) was detected in only Regions 1 and 3. In Region 1, its prevalence among PL, farm animals, and broodstock was 20%, 45%, and 10%, respectively. It was not present in wild *P. vannamei* but was detected in wild *P. stylirostris*, with a 13% prevalence. In Region 3, it was present in PL and broodstock, with 33 and 90% prevalence, respectively. It was not detected in wild *P. monodon* pathogens such as CMNV, TSV, and DIV1.

The following pathogens were not detected in any region: MrBdv, PvNV, IMNV, YHV, MrNV, and XSV. A comparison of the most prevalent pathogens in this study with those found during the first survey revealed that WzSV8 had a high prevalence (>80%) during the year of the study and that the prevalence of almost all pathogens increased in the last six months, with special attention given to EHP (Table 7).

### 3.2. Histopathology Results

Approximately 150 samples were analyzed for histopathology in the second part of this survey. In general, coinfections that consisted of WzSV8, DHPV, chronic midgut inflammation, and tubule distension/epithelial atrophy consistent with Pir A/B toxicity or other bacteria were the most common pathologies in all regions (Table 8).

Wenzhou shrimp virus 8 (WzSV8) viral inclusions (VINs) are basophilic structures located within vacuoles in the cytoplasm of hepatopancreatic (HP) tubule cells and predominantly appear circular in cross-sections. In contrast, Decapod hepanhamaparvovirus (DHPV) VINs are found in the nuclei, exhibit a magenta coloration, are ovoid in shape, and are separated from marginated chromatin by an unstained space. Occasionally, smaller DHPV inclusions may displace the nucleolus, creating a crescent-shaped appearance. Co-infection of the HP with both DHPV and WzSV8 is frequently observed (Figure 1).

Histological analysis employing a modified methyl green-pyronin stain, designed to differentiate DNA and RNA, revealed that both DHPV and WzSV8 appear to infect not only the hepatopancreas but also the ceca and intestine (Figure 1). Notably, WzSV8 displays unique lighter double inclusions (LDIs) [27], which are considered pathognomonic for the virus. These LDIs are characterized by additional, smaller eosinophilic inclusions adjacent to the primary, circular basophilic inclusions within vacuoles. In some specimens, sloughed, rounded WzSV8-infected cells were observed within the lumen of the HP tubules.

Interestingly, WzSV8 VINs, including both basophilic and LDI types, were frequently detected in the ovaries of certain specimens, at both developing and mature stages, as previously reported [1]. This finding suggests a broader tissue tropism for WzSV8 beyond the hepatopancreas. WzSV8 VINs were detected in PL, farm animals, broodstock, and wild animals in all three regions. Their prevalence ranged from 9% in larvae in Regions 1 and 4 to 100% in farm animals in Region 4 (Table 8). DHPV VINs were never found in PL, and they were not found either in any sample from Region 3. Region 1 included farm animals, broodstock, and wild *P. vannamei*, but not *P. stylirostris*.

After Wenzhou shrimp virus 8 (WzSV8), the most significant disorder observed in the hepatopancreas was chronic midgut inflammation accompanied by tubule distension and epithelial atrophy, consistent with PirA/B toxin or other bacterial toxicity (Table 8, Figure 2). Lesions in the hepatopancreas were evident across all sizes of shrimp, from post-larvae (PL) to broodstock. Notably, as shown in Table 8, the severity and size of these lesions increased with shrimp size. Interestingly, these lesions were also detected in all species of wild *Penaeus* sampled in Regions 1 and 3.

Baculovirus penaei (BP), also referred to as singly enveloped nucleopolyhedrovirus (*PvSNPV*), was sporadically observed in the hepatopancreas of broodstock. Specifically, BP was detected in 3 out of 196 animals in Region 1 and in 5 out of 10 animals in Region 3 (Figure 2).

Additionally, hemocytic enteritis, characterized by inflammation of the intestinal walls with hemocyte infiltration, was commonly observed across all shrimp sizes in Region 1 and in farmed animals in Region 4. In Region 1, this condition was consistently associated with muscle necrosis and the presence of bacterial aggregates accompanied by hemocyte infiltration. In some cases, bacterial infections were further complicated by melanized reactions in the cuticle of the exoskeleton (Figure 3).

Lymphoid organ spheroids (LOSs) were never present in PL and wild *P. monodon* but could be found in farms, broodstock, and in wild *P. vannamei* and *P. stylirostris* (Table 8 and Figure 4). In contrast to Region 3, Regions 1 and 4 had the highest prevalence. Interestingly, LOS was also found in wild *P. vannamei* and *P. stylirostris*.

Histopathology of the PLs (Regions 1 and 4) revealed sloughing cells and slight hemocyte infiltration with a clear bacterial infection in the hepatopancreatic tubules (Figure 5). WzSV8 VIN was also detected (Figure 5E,F).

Lesions associated with white spot syndrome virus (WSSV) were only observed in Region 1, specifically in farmed shrimp and broodstock. Although the prevalence was low, the severity of the lesions was high, with lesion grades exceeding 3 (Table 8). WSSV viral inclusions (VIN) were detected in various tissues, including the stomach, gills, hematopoietic tissue, antennal gland, epidermis, and stomach (Figure 6A).

Infectious hypodermal and hematopoietic necrosis virus (IHHNV) was identified in 1 single sample from farmed shrimp out of 64 samples in Region 1. Cowdry type A inclusions, which are characteristic of IHHNV, were observed in the rostrum, pleopods, and nervous tissue (Figure 6B).

Clusters of microsporidian spores were primarily observed within hepatopancreatic cells, striated muscle sarcomeres, and antennal glands (Figure 7). Histopathological lesions caused by microsporidia were detected exclusively in Region 1 (Table 8).

Gregarines and nematodes of various sizes, including those found in wild *P. vannamei* and *P. stylirostris*, were observed in Region 1. These parasites were only detected in farmed animals in Region 3 and were absent in animals from Region 4. Larval nematodes (Figure 8A,B) were commonly found encysted in the foregut wall, near the junction with the midgut. Gregarine gametocytes were located within the lumen of the posterior midgut cecum, while trophozoites were found in the lumen of both the anterior midgut cecum and the midgut intestine (Figure 8C,D). Ciliate protozoans were observed in the gills, accompanied by an inflammatory response and melanized reactions (Figure 8E,F).

## 4. Discussion

The present study represents the second phase of a year-long investigation into histopathological lesions and the identification of pathogens in *P. vannamei* samples collected from hatcheries, farms, and maturation units across three distinct regions of Latin America. Additionally, samples from wild *P. vannamei*, *P. stylirostris*, and *P. monodon* were included in the analysis. This study compares multiple primer sets targeting various pathogens to assess geographical variations and specificity. A key observation from the study was the variation in co-infection levels across the three regions, which appear to be influenced by the culture system and the health status of the broodstock/source of the postlarvae. *Wenzhou shrimp virus 8* (WzSV8) emerged as the most prevalent pathogen detected in all sampled regions, both by PCR and histology, irrespective of shrimp size or environment, including wild penaeid samples from both this and the previous study [1].

WzSV8 viral inclusion bodies (VINs) were seldom found in isolation and were often associated with midgut lesions, which were presumed to be caused by bacterial infections. A lower prevalence of Decapod hepatopancreatic parvovirus (DHPV), *White spot syndrome virus* (WSSV), and Baculovirus penaei (BP) was also observed. The consistent presence of WzSV8 in both farmed and wild animals across regions strongly suggests that it is endemic to the area and may be globally distributed.

Midgut histopathological findings in this study were suggestive of bacterial infections, including septic hepatopancreatic necrosis (SHPN), chronic AHPND, and/or rickettsia-like organisms (RLO). However, due to the lack of in situ hybridization capabilities, we were unable to definitively link these lesions to specific pathogens or digestive system alterations. Therefore, we could not confirm whether the pathogens identified in the lesions matched those detected by PCR in samples from different shrimp of the same source. Consistent with previous findings [1], the high prevalence of PCR-positive samples for *Vibrio* species, rickettsia-like bacteria (RLB), and AHPND, along with the histopathological lesions observed in the hepatopancreas and intestine (with bacterial involvement), strongly suggests a potential association between these bacterial pathogens and the observed lesions. These results further support the hypothesis that *Vibrio* species and related bacterial diseases are prevalent in the region [1].

In Region 1, located in the Pacific Ocean, biosecurity measures were minimal, and there was no rigorous selection process for broodstock; animals were sourced from farms without considering their health status. In contrast, Region 3, situated in the Atlantic Ocean, comprises an open farm with no biosecurity protocols but uses specific pathogen-free (SPF) broodstock and larvae. Similarly, Region 4 is a superintensive farm with lined ponds in the Pacific Ocean, utilizing well water and employing SPF broodstock and larvae.

Histopathological lesions in postlarvae (PLs) from Regions 1 and 4 were primarily bacterial in origin. The hepatopancreatic tubules exhibited sloughing cells and mild hemocyte infiltration, indicating a bacterial infection. WzSV8 viral inclusion bodies (VINs) were also observed (Figure 5). High and sudden mortalities of *P. vannamei* PLs were reported in Latin American hatcheries [55], attributed to a *Vibrio* species carrying the VpPirAB genes. Histopathological examination revealed extensive sloughing and detachment of hepatopancreocytes. This condition was tentatively named postlarval acute hepatopancreatic necrosis disease (PL-AHPND) to differentiate it from other pathologies affecting postlarval shrimp [55].

It is noteworthy that pathogens such as *Spiroplasma* and EHP were detected by PCR but not histopathologically. Specifically, the presence of *Spiroplasma* species, particularly *S. mirum*, in farmed animals from Region 3 (Table 4) warrants further investigation. For EHP, there was an observed increase in prevalence from the first survey [55] to the present study. Although further factors need to be determined, EHP, in conjunction with opportunistic bacteria, *Propionigenium*, and other unknown stress factors, may be a primary contributor to a condition referred to as white feces syndrome, which poses a significant threat to *P. vannamei* culture [56].

In penaeid species, the lymphoid organ plays a crucial role in immune defense and is potentially the major phagocytic organ [57]. It is also a key site for viral degradation within lymphoid organ spheroids (LOSs) [57]. LOSs were observed in all regions, across all shrimp sizes, and in both wild *P. vannamei* and *P. stylirostris*, except for PLs and *P. monodon* (Table 8). The extensive LOS formation, as reported previously [1], may be related to the tolerance of penaeid prawns to bacterial and viral infections.

PCR analysis indicated that 26% of the *IHHNV* cases were associated with endogenous viral elements (EVE), while 23% were likely attributed to infectious IHHNV (Table 6). It is noteworthy that only one sample from 69 (from farm animals in Region 1) exhibited pathognomonic Cowdry type A inclusion (CAI) lesions (Figure 6B), although these lesions were found in different specimens from the same source. When using multiple PCR primer sets of equal sensitivity targeting different regions of the virus genome, it is essential that all primer pairs yield the expected amplicons. Failure to achieve this suggests that the amplicons detected likely arise from EVEs rather than complete viral genomes [58,59]. IHHNV has been shown to integrate into host genomic DNA and does not typically correlate with histological lesions associated with IHHNV infection. A study by [60] demonstrated that a separate scrambled cluster of genome fragments from an extant type of IHHNV could be integrated into the shrimp genome, and false positive PCR results may occur using detection methods recommended by the World Organization for Animal Health (WOAH).

The presence of EVEs derived from IHHNV and other viruses emphasizes the need for caution, especially when reporting new viral findings from previously untested locations [61,62].

More than 20 species of *Microsporidia*, belonging to 17 genera, have been reported in various decapod species from the families *Penaeidae* and *Caridea* [36]. The two most prevalent non-EHP microsporidian parasites in penaeids are *Agmasoma penaei* and *Perezia nelsoni*, which differ in life cycle, morphology, and tissue tropism. Pasharawipas et al. [22,23] developed a specific DNA probe to identify *Agmasoma* as the causative agent of muscle infections in *P. merguensis* and *P. monodon* in Thailand. Later, Sokolova et al. [37] suggested that muscle infections previously attributed to *A. penaei* may, in fact, be due to concurrent infections with *P. nelsoni*, which primarily infects the gonads, while *A. penaei* affects the muscles. In the present study, the PCR prevalence of microsporidia averaged 85% across the three regions; however, histopathological analysis revealed that fewer than 2% of the samples showed evidence of microsporidian infections. These infections were primarily found in the muscles and hepatopancreas, with a notable restriction to farmed animals and broodstock from Region 1. Further investigations are needed to better understand the dynamics and significance of this group of parasites.

A high prevalence of gregarines and nematodes was observed, particularly in Regions 1 and 3, throughout the study period. In Region 1, these parasites were present in all shrimp sizes, including wild *P. vannamei* and *P. stylirostris*, with a prevalence of 100% (Table 8). In contrast, they were absent in Region 4, which may be linked to the use of lined ponds. In the wild, penaeids serve as both intermediate and final hosts for a variety of parasites, typically acquiring infections through the ingestion of free-living stages or infected planktonic hosts [63,64]. The presence of gregarines, microsporidia, trematodes, cestodes, and nematodes in wild penaeids has been well-documented over the past six decades [65,66,67]. This highlights the potential role of parasites in introducing and disseminating pathogens within the ecosystem, even among seemingly healthy individuals.

Throughout the year, it was observed that animals, despite appearing healthy, often harbored multiple pathogens, including viruses, bacteria, and metazoan parasites. While the presence of a pathogenic agent is generally necessary for disease onset, it is not always sufficient on its own [68]. A variety of factors, such as stocking density and environmental conditions, can compromise the host’s immune system, increasing the likelihood of disease development [69,70].

In shrimp culture, it is common to attribute disease to a single pathogen; however, multiple infections often occur simultaneously, with the combined effect of these infections sometimes being more severe than when a single pathogen is involved [71]. Crustaceans, whether from farm or wild environments, can host a diverse array of DNA and RNA viruses [72], and the health outcomes of these organisms may be influenced by the contributions of multiple pathogens. This complexity is often overlooked in diagnostic practices, which typically focus on individual agents [73,74].

Coinfections not only complicate disease dynamics but also have the potential to drive genetic exchange between pathogens, leading to recombinant viruses that can impact viral evolution and disease progression, ultimately affecting the host’s survival [73,74]. In some cases, coinfections can also reduce or exacerbate disease severity. When cells are coinfected, one virus can influence the replication of another, a phenomenon known as viral interference. Such interactions have been documented between IHHNV and WSSV [75], as well as between TSV and YHV [76,77].

Moreover, mutualistic viruses can benefit the host by supplying new genes or inducing epigenetic changes in the host genome that lead to beneficial outcomes [78]. This tolerance, however, does not equate to resistance, as shrimp may still become infected and potentially develop disease when a triggering factor is introduced [78].

The importance of selecting the appropriate PCR primers specific to target pathogens cannot be overstated, especially given the variation in pathogens’ genetic diversity across different geographic locations. This genetic diversity, influenced by host distribution, environmental factors, and viral adaptations, may explain the inconsistencies observed in several PCR primers [79]. In this new era of advanced molecular biology and artificial intelligence, histology, along with the careful selection of appropriate primers, remains essential for accurate pathogen identification and disease management, ensuring that diagnostic tools keep pace with evolving scientific advancements.

In conclusion, this study emphasizes several key considerations for disease management in shrimp populations:**Geographic Distribution and Pathogen Prevalence:** Understanding the regional variability of pathogen prevalence is essential for effective disease control.**PCR Primer Selection:** The critical importance of using geographically specific primers is highlighted, as incorrect primer selection can lead to false positives due to primer cross-reactivity or environmental DNA contamination. These factors should be considered when developing diagnostic protocols for pathogen detection.

## Figures and Tables

**Figure 1 viruses-17-00187-f001:**
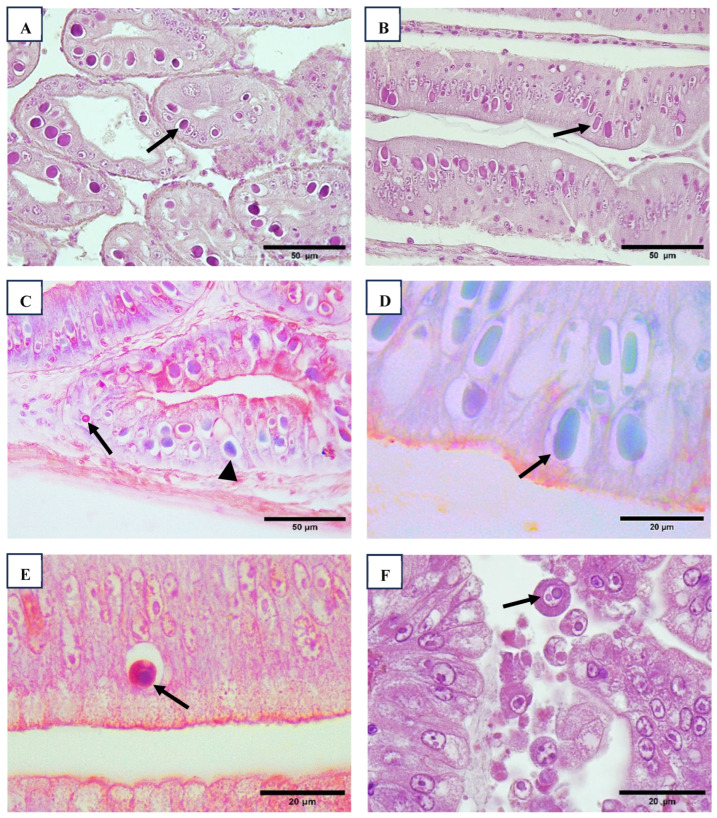
HPV/DHPV inclusion bodies in the hepatopancreatic cells (arrow) ((**A**), 40×). Anterior caecum cells infected with HPV/DHPV (arrow) ((**B**), 40×). H&E stain. Anterior caecum cells infected with WzSV8 (arrow) and HPV/DHPV (arrowhead) ((**C**), 40×). Methyl green-pyronin modified stain to distinguish DNA and RNA. Intestine cells infected with HPV/DHPV (arrow) ((**D**), 100×). Methyl green-pyronin modified stain to distinguish DNA and RNA. Ceca infected with WzSV8 (arrow) ((**E**), 100×). Methyl green-pyronin modified stain to distinguish DNA and RNA. Hepatopancreatic cells with inclusion bodies of WZV-8, ((**F**), 100×). H&E stain.

**Figure 2 viruses-17-00187-f002:**
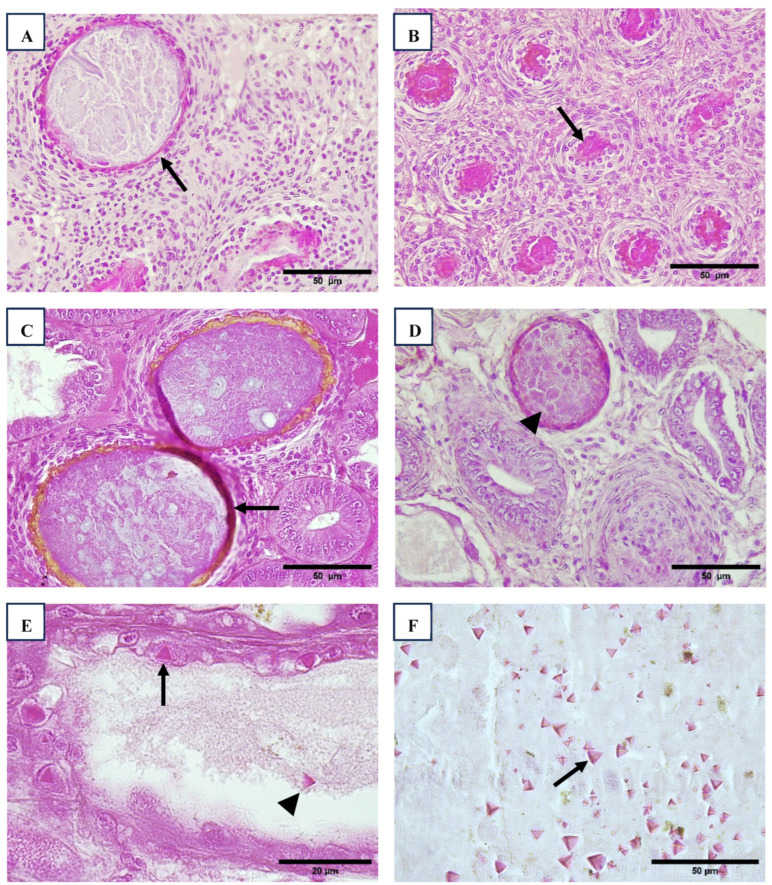
Histological samples of hepatopancreas with presence of hemocytes, melanized and necrotic tubules (arrow), intracellular bacteria (arrowhead) ((**A**–**D**), 40×). Hepatopancreatic cells infected with occlusion body of BP-PvSNPV (arrow) and free tetrahedral body in the lumen (arrowhead) ((**E**), 100×). Tetrahedral BP-body of PvSNPV in intestine lumen (arrow) ((**F**), 40×). H&E stain.

**Figure 3 viruses-17-00187-f003:**
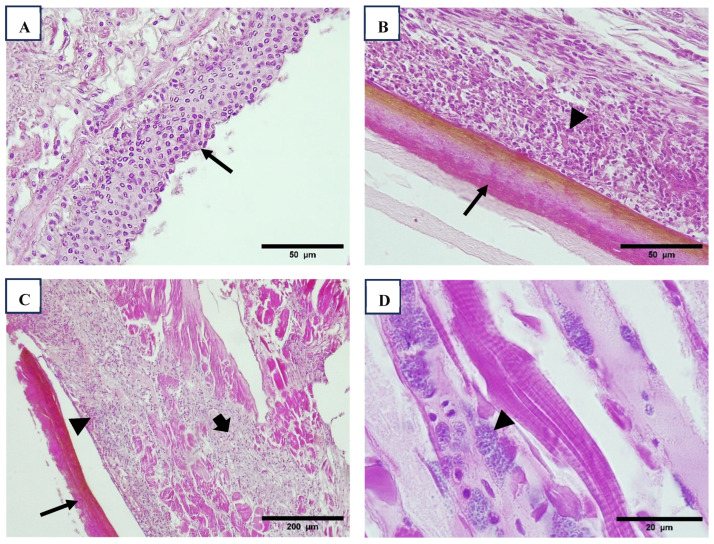
Cross-section views intestinal epithelium with enteritis, note the inflammation of intestine walls with hemocyte infiltration (arrow) ((**A**), 40×). Melanized reaction in cuticle of exoskeleton (arrow), infiltrated reaction in epidermis (arrowhead) ((**B**), 40×, (**C**), 10×) and muscle necrosis (thick arrow). Muscle necrosis and presence of pack of bacteria ((**D**), 100×). H&E stain.

**Figure 4 viruses-17-00187-f004:**
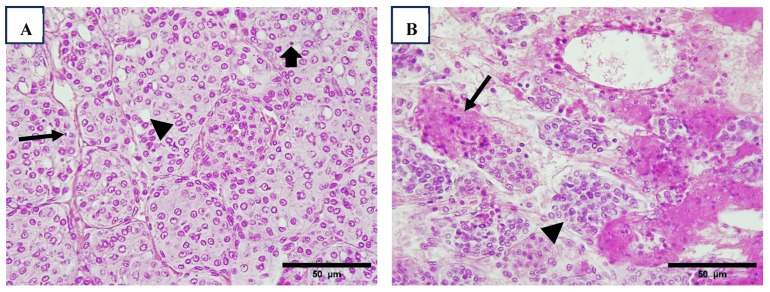
Presence of LOS in lymphoid organ, note the different cellular reaction in LOS, pycnotic nuclei (arrow), vacuoles (arrowhead), and RNA-like virus (thick arrow) ((**A**), 40×). Lymphoid organ with melanized reaction (arrow) and LOS (arrowhead) ((**B**), 40×). H&E stain.

**Figure 5 viruses-17-00187-f005:**
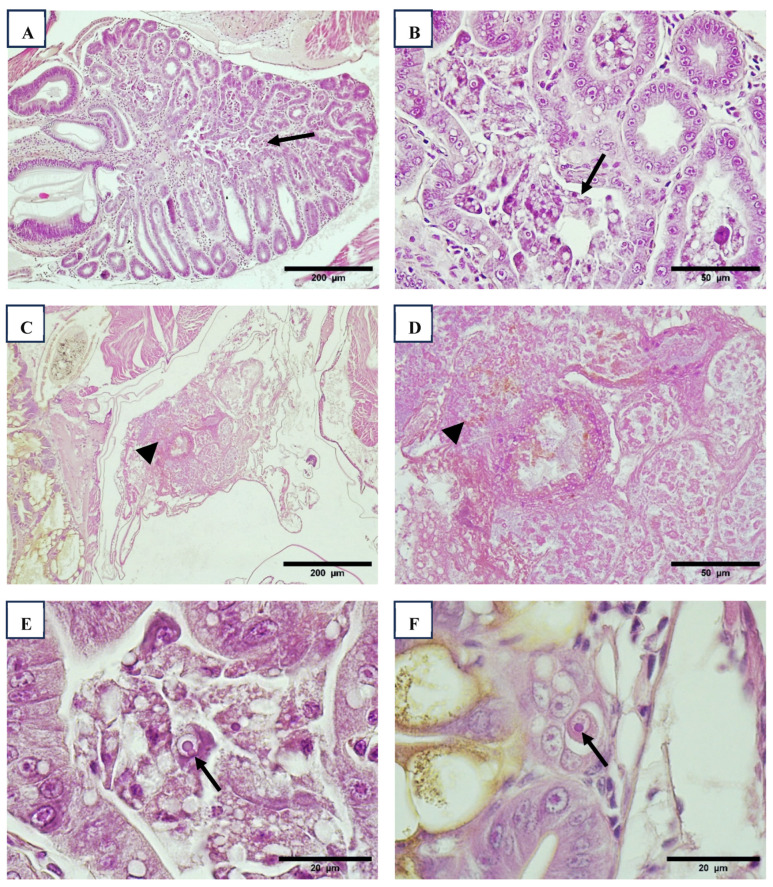
Hepatopancreas of *P. vannamei* larvae with sloughing cells and slight hemocyte infiltration (arrow) ((**A**), 10×. (**B**), 40×) and bacterial infection in hepatopancreatic tubules (arrowhead) ((**C**), 10×. (**D**), 40×). Hepatopancreatic cell and sloughing cells of hepatopancreas with WZV-8 inclusion body (arrow) ((**E**,**F**), 100×). H&E stain.

**Figure 6 viruses-17-00187-f006:**
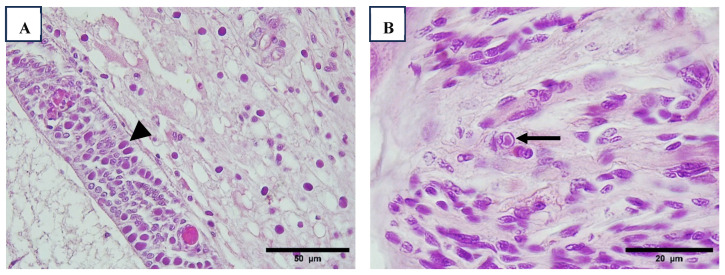
Histological sections of epithelial of stomach with WSSV (arrowhead) ((**A**), 40×). Histological section of nerve cordon with nervous cell presented inclusion body Cowdry type A typical of IHHNV (arrow) ((**B**), 100×). H&E stain.

**Figure 7 viruses-17-00187-f007:**
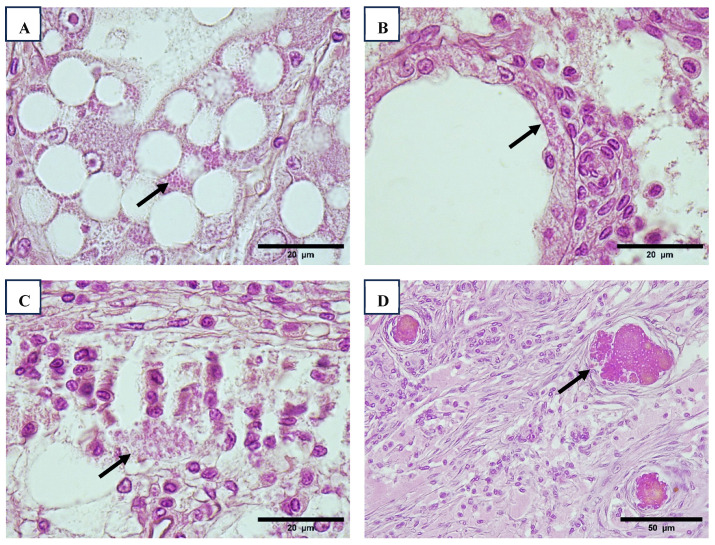
Infection by microsporidian in hepatopancreatic cells (arrow) ((**A**), 100×). Microsporidia was identified packed in antennal gland (**B**) and connective tissue of stomach (arrow) ((**C**), 100×); both tissues have a slight hemocyte infiltration. Shrimp nephrocomplex with sign of damage in the antennal gland, observing hemocyte infiltration and tubules within melanized reaction and melanized encapsulations (arrow) ((**D**), 40×). H&E stain.

**Figure 8 viruses-17-00187-f008:**
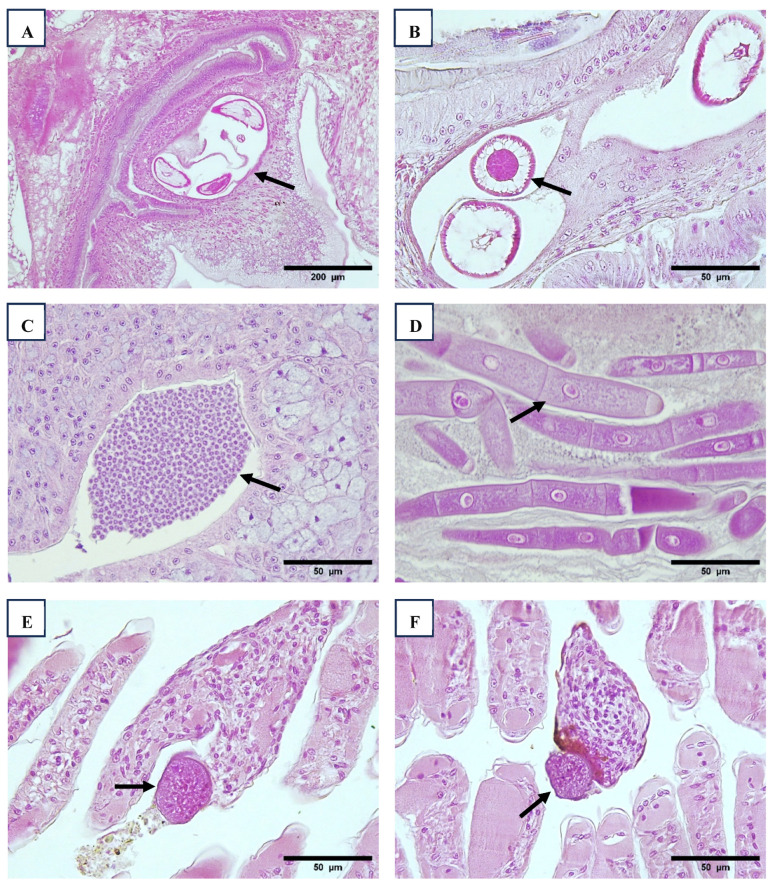
Histological sections of anterior digestive part with presence of nematode cyst encysted in connective tissue of stomach or anterior caecum and surrounded by hemocytes (arrow) ((**A**), 10×. (**B**), 40×). Presence of gregarine cyst in tegumental gland of posterior intestine folds ((**C**), 40×). Trophozoite stage of gregarines in intestine lumen ((**D**), 40×). Presence of ciliate protozoans in the gills with hemocyte infiltration or melanized reaction (arrow) ((**E**,**F**), 40×). H&E stain.

**Table 1 viruses-17-00187-t001:** Comparison of prevalence using 3 different sets of primers for WzSV8 in *P. vannamei*, *P. stylirostris*, and *P. monodon* in samples from different regions of Latin America.

	WzSV8 ^1^ 504F/R 170F/R 540 bp 170 bp	WzSV8 ^2^ 428F/R 168F/R 428 bp 168 bp	PvSV ^3^ 3136F/3268R 133 bp
**Region #1**			
PL	3/5 (60%)	5/5 (100%)	4/5 (80%)
Farm	52/58 (90%)	57/58 (98%)	55/58 (95%)
Broodstock	23/29 (79%)	26/29 (90%)	25/29 (86%)
Wild *P. vannamei*	3/10 (30%)	0/10 (0%)	2/10 (20%)
Wild *P. stylirostris*	3/8 (38%)	7/8 (88%)	8/8 (100%)
**Region #3**			
PL	3/3 (100%)	2/3 (67%)	2/3 (67%)
Farm	5/8 (63%)	8/8 (100%)	8/8 (100%)
Broodstock	0/10 (0%)	3/10 (30%)	1/10 (10%)
Wild *P. monodon*	0/4 (0%)	1/4 (25%)	3/4 (75%)
**Region #4**			
PL	6/7 (86%)	7/7 (100%)	7/7 (100%)
Farm	4/5 (80%)	5/5 (100%)	5/5 (100%)
Broodstock	1/1 (100%)	1/1 (100%)	1/1 (100%)
**Average ****	**103/148 (70%)**	**122/148 (82%)**	**121/148 (82%)**

****** Average pool of all samples in all regions. ^1^ [26] ^2^ [27] ^3^ [7].

**Table 2 viruses-17-00187-t002:** Comparison of sensitivity using 4 different sets of primers for Hepanhamaparvovirus (DHPV) and 1 for Macrobrachium Bidnavirus (MrBdv) in *P. vannamei*, *P. stylirostris*, and *P. monodon* in samples from different regions of Latin America.

	H441F/R ^1^ HPVnF/R	H441F/R ^2^ HPVnF/R	1120F/R 592 bp ^3^	DHPV-U 1538 F DHPV-U 1887R DHPV-U 1622F 350 bp 256 bp ^4^	MrBdv L/R 392 bp ^5^
**Region #1**					
PL	0/5 (0%)	0/5 (0%)	0/5 (0%)	0/5 (0%))	0/5 (0%)
Farm	23/58 (40%)	24/58 (41%)	4/58 (7%)	8/58 (14%)	0/58 (0%)
Broodstock	4/26 (15%)	4/26 (15%)	0/26 (0%)	0/26 (0%)	0/26 (0%)
Wild *P.* *vannamei*	0/10 (0%)	0/10 (0%)	0/10 (0%)	0/10 (0%)	0/10 (0%)
Wild *P. stylirostris*	0/8 (0%)	0/8 (0%)	0/8 (0%)	0/8 (0%)	0/8 (0%)
**Region #3**					
PL	1/3 (33%)	0/3 (0%)	0/3 (0%)	0/3 (0%)	0/3 (0%)
Farm	0/8 (0%)	0/8 (0%)	0/8 (0%)	0/8 (0%)	0/8 (0%)
Broodstock	0/10 (0%)	0/10 (0%)	0/10 (0%)	0/10 (0%)	0/10 (0%)
Wild *P.* *monodon*	0/4 (0%)	0/4 (0%)	0/4 (0%)	0/4 (0%)	0/4 (0%)
**Region #4**					
PL	0/7 (0%)	0/7 (0%)	0/7 (0%)	0/7 (0%)	0/7 (0%)
Farm	2/5 (40%)	1/5 (20%)	0/5 (0%)	0/5 (0%)	0/5 (0%)
Broodstock	1/1 (100%)	0/1 (0%)	0/1 (0%)	0/1 (0%)	0/1 (0%)
**Average ****	**31/145 (21%)**	**29/145 (20%)**	**4/145 (3%)**	**8/145 (6%)**	**0/145 (0%)**

****** Average pool of all samples in all regions. ^1^ [28] ^2^ [29] ^3^ [30] ^4^ [31] ^5^ [32].

**Table 3 viruses-17-00187-t003:** Comparison of prevalence using 3 different primers for RLB/NHPB in *P. vannamei*, *P. stylirostris*, and *P. monodon* in samples from different regions of Latin America.

	Bact F/R 1500 bp Rik F/R532 bp ^1^	Rp877p Rp1258 n 382 bp ^2^	NHPF2 NHPR2 379 bp ^3^
**Region #1**			
PL	0/5 (0%)	1/5 (20%)	0/5 (0%)
Farm	1/58 (2%)	17/58 (29%)	0/66 (0%)
Broodstock	17/41 (41%)	21/41 (51%)	0/26 (0%)
Wild *P. vannamei*	0/10 (0%)	0/10 (0%)	0/10 (0%)
Wild *P. stylirostris*	0/8 (0%)	0/8 (0%)	0/8 (0%)
**Region #3**			
PL	0/3 (0%)	0/3 (0%)	0/3 (0%)
Farm	0/8 (0%)	5/8 (63%)	0/8 (0%)
Broodstock	0/10 (0%)	0/10 (0%)	0/10 (0%)
Wild *P. monodon*	0/4 (0%)	0/4 (0%)	0/4 (0%)
**Region #4**			
PL	0/7 (0%)	1/7 (0%)	0/7 (0%)
Farm	0/5 (0%)	0/5 (0%)	0/5 (0%)
Broodstock	0/1 (0%)	0/1 (0%)	0/1 (0%)
**Average ****	**18/160 (11%)**	**45/160 (28%)**	**0/153 (0%)**

****** Average pool of all samples in all regions. ^1^ [17] ^2^ [33] ^3^ [19].

**Table 4 viruses-17-00187-t004:** Comparison of sensitivity using 3 different sets of primers for *Spiroplasma* in *P. vannamei*, *P. stylirostris* and *P. monodon* in samples from different regions of Latin America.

	CSF/R 269 bp ^1^	F28/R5 271 bp ^2^	Pri-1/2 1200 bp ^3^
**Region #1**			
PL	1/5 (20%)	0/5 (0%)	0/5 (0%)
Farm	28/62 (45%)	1/62 (2%)	6/62 (10%)
Broodstock	0/26 (0%)	2/26 (8%)	0/26 (0%)
Wild *P. vannamei*	0/10 (0%)	0/10 (0%)	0/10 (0%)
Wild *P. stylirostris*	0/8 (0%)	0/8 (0%)	0/8 (0%)
**Region #3**			
PL	0/3 (0%)	0/3 (0%)	0/3 (0%)
Farm	1/8 (13%)	0/8 (0%)	8/8 (100%)
Broodstock	0/10 (0%)	0/10 (0%)	0/10 (0%)
Wild *P. monodon*	0/4 (0%)	0/4 (0%)	0/4 (0%)
**Region #4**			
PL	0/7 (0%)	0/7 (0%)	0/7 (0%)
Farm	0/5 (0%)	0/5 (0%)	0/5 (0%)
Broodstock	0/1 (0%)	0/1 (0%)	0/1 (0%)
**Average ****	**30/149 (20%)**	**3/149 (2%)**	**14/149 (9%)**

****** Average pool of all samples in all regions. ^1^ [16] ^2^ [34] ^3^ [35].

**Table 5 viruses-17-00187-t005:** Comparison of the prevalence of Microsporidia in *P. vannamei*, *P. stylirostris*, and *P. monodon* using 3 different primers in samples from different regions of Latin America.

	TS1/TS2 790 bp ^1^	18f/1492r 1200 bp ^2^	Ss2 18f3/ ss1492r ^3^
**Region #1**			
PL	3/5 (60%)	0/5 (0%)	0/5 (0%)
Farm	46/57 (81%)	16/57 (28%)	31/57 (54%)
Broodstock	20/26 (77%)	6/26 (23%)	8/26 (31%)
Wild *P. vannamei*	9/10 (90%)	2/10 (20%)	2/10 (20%)
Wild *P. stylirostris*	7/8 (88%)	0/8 (0%)	6/8 (75%)
**Region #3**			
PL	1/3 (33%)	0/3 (0%)	0/3 (0%)
Farm	8/8 (100%)	2/8 (25%)	4/8 (50%)
Broodstock	8/10 (80%)	0/10 (0%)	6/10 (60%)
Wild *P. monodon*	3/4 (75%)	0/4 (0%)	3/4 (75%)
**Region #4**			
PL	1/7 (14%)	0/7 (0%)	1/7 (14%)
Farm	5/5 (100%)	4/5 (80%)	5/5 (100%)
Broodstock	1/1 (100%)	0/1 (0%)	1/1 (100%)
**Average ****	**112/144 (85%)**	**30/144 (21%)**	**67/144 (47%)**

****** Average pool of all samples in all regions. ^1^ [22] ^2^ [36] ^3^ [38].

**Table 6 viruses-17-00187-t006:** Comparison of prevalence using 4 different primers for IHHNV in *P. vannamei*, *P. stylirostris*, and *P. monodon* in samples from different regions of Latin America.

	309 F/R 309 bp ^1^	392 F/R 392 bp ^2^	389 F/R 389 bp ^3^	77012F/77353R (356 bp) ^4^	IHHNV ***	Eve ***
**Region #1**						
PL	3/5 (60%)	3/5 (60%)	3/5 (60%)	0/5 (0%)	0/5 (0%)	3/5 (60%)
Farm	44/58 (76%)	38/58 (66%)	37/58 (64%)	26/58 (45%)	26/58 (45%)	19/58 (33%)
Broodstock	6/26 (23%)	8/26 (31%)	6/26 (21%)	4/26 (15%)	4/26 (15%)	4/26 (15%)
Wild *P. vannamei*	8/10 (80%)	7/10 (70%)	5/10 (50%)	3/10 (30%)	3/10 (30%)	5/10 (50%)
Wild *P. stylirostris*	0/8 (0%)	0/8 (0%)	0/8 (0%)	0/8 (0%)	0/8 (0%)	0/8 (0%)
**Region #3**						
PL	0/3 (0%)	0/3 (0%)	1/3 (33%)	0/3 (0%)	0/3 (0%)	1/3 (33%)
Farm	0/8 (0%)	0/8 (0%)	0/8 (0%)	0/8 (0%)	0/8 (0%)	0/8 (0%)
Broodstock	0/10 (0%)	0/10 (0%)	0/10 (0%)	0/10 (0%)	0/10 (0%)	0/10 (0%)
Wild *P. monodon*	0/4 (0%)	1/4 (25%)	2/4 (50%)	1/4 (25%)	0/4 (0%)	2/4 (50%)
**Region #4**						
PL	3/7 (43%)	0/7 (0%)	1/7 (14%)	0/7 (0%)	0/7 (0%)	3/7 (43%)
Farm	1/5 (20%)	1/5 (20%)	1/5 (20%)	1/5 (20%)	1/5 (20%)	0/5 (0%)
Broodstock	0/1 (0%)	0/1 (0%)	0/1 (0%)	0/1 (0%)	0/1 (0%)	0/1 (0%)
**Average ****	**65/145 (45%)**	**58/145 (40%)**	**56/145 (39%)**	**35/145 (24%)**	**34/145 (23%)**	**37/145 (26%)**

****** Average pool of all samples in all regions. *** When tested using multiple PCR primers, all primers must be present for a functional virus to replicate. However, if some but not all primers amplify, this pattern is a clear indication that the PCR primers are measuring an eve (endogenous viral element) and not the complete virus. ^1^ [53] ^2^ [39] ^3^ [40] ^4^ [41].

**Table 7 viruses-17-00187-t007:** Comparison of prevalence using 6 different primers for EHP in *P. vannamei*, *P. stylirostris*, and *P. monodon* in samples from different regions of Latin America.

	EHP 510 bp ^1^	EHP900– 1000 bp ^2^	VE-EHP-SWP-365 bp ^3^	EHP B-tubulin 262 bp ^3^	SSU ENF779 F1 SSU ENF779 R1 SSU ENF176 F1 SSU ENF176 R1 779 bp 176 bp ^4^	SWP_1F SWP_1R SWP_2F SWP_2R 514 bp 148 bp ^5^
**Region #1**						
PL	0/5 (0%)	0/5 (0%)	0/5 (0%)	0/5 (0%)	0/5 (0%)	0/5 (0%)
Farm	20/72 (28%)	10/72 (14%)	1/72 (1%)	3/72 (4%)	30/72 (42%)	19/72 (26%)
Broodstock	8/29 (28%)	1/29 (3%)	0/29 (0%)	5/29 (17%)	8/29 (28%)	10/29 (34%)
Wild *P. vannamei*	0/10 (0%)	0/10 (0%)	0/10 (0%)	0/10 (0%)	0/10 (0%)	0/10 (0%)
Wild *P. stylirostris*	0/8 (0%)	0/8 (0%)	0/8 (0%)	0/8 (0%)	1/8 (13%)	0/8 (0%)
**Region #3**						
PL	0/3 (0%)	0/3 (0%)	0/3 (0%)	0/3 (0%)	0/3 (0%)	0/3 (0%)
Farm	0/8 (0%)	5/8 (0%)	0/8 (0%)	0/8 (0%)	0/8 (0%)	0/8 (0%)
Broodstock	0/10 (0%)	0/10 (0%)	0/10 (0%)	0/10 (0%)	0/10 (0%)	0/10 (0%)
Wild *P. monodon*	0/4 (0%)	0/4 (0%)	0/4 (0%)	0/4 (0%)	0/4 (0%)	0/4 (0%)
**Region #4**						
PL	0/7 (0%)	0/7 (0%)	0/7 (0%)	0/7 (0%)	1/7 (14%)	0/7 (0%)
Farm	0/5 (0%)	0/5 (0%)	0/5 (0%)	0/5 (0%)	5/5 (100%)	0/5 (0%)
Broodstock	0/1 (0%)	0/1 (0%)	0/1 (0%)	0/1 (0%)	1/1 (100%)	0/1 (0%)
**Average ****	**28/162 (17%)**	**16/162 (10%)**	**1/162 (1%)**	**8/162 (5%)**	**46/162 (28%)**	**29/162 (18%)**

****** Average pool of all samples in all regions. ^1^ [43] ^2^ [21] ^3^ [44] ^4^ [42] ^5^ [45].

**Table 8 viruses-17-00187-t008:** Histological average of major lesions in the 3 regions.

			Hepatop ^1^		Intestine ^2^		Muscle ^3^		LO ^4^	
	Ave (g) ^10^	n ^11^	% ^12^	Grade ^13^	% ^12^	Grade ^13^	% ^12^	Grade ^13^	% ^12^	Grade ^13^
**Region 1**										
PL	ND **^14^**	250	1.7	2.2	0.3	1.0	0.7	1.0	-	-
Farm	13.6	320	13.9	1.4	4.1	1.4	6.4	1.2	54.1	1.7
Broodstock	33.8	196	17.8	1.4	5.0	1.6	7.0	1.1	50.8	1.7
Wild *P. vannamei*	35.0	10	19.8	1.2	12.5	1.0	20.5	1.0	87.5	2.3
Wild *P. stylirostris*	39.8	8	25.0	1.0	26.7	1.0	30.0	8.9	26.7	1.0
**Region 3**										
PL	ND **^14^**	150	-	-	-	-	-	-	-	-
Farm	14.4	24	9.2	1.2	-	-	-	-	4.1	1.0
Broodstock	50.6	10	58.0	1.8	-	-	95.0	2.8	10.0	1.0
Wild *P. monodon*	66.9	5	8.0	0.4	-	-	-	-	-	-
**Region 4**										
PL	ND **^14^**	330	1.6	3.8	-	-	-	-	-	-
Farm	10.7	27	29.1	1.5	7.3	1.1	40.0	2.1	50.0	1.0
Broodstock	66	2	62.5	1.0	-	-	50.0	1.0	50.0	1.0
	**WSSV** **^5^**		**DHPV** **^6^**		**WZ8V** **^7^**		**Microspor** **^8^**		**Grega/Nemat** **^9^**	
	**%** **^12^**	**Grade** **^13^**	**%** **^12^**	**Grade** **^13^**	**%** **^12^**	**Grade** **^13^**	**%** **^12^**	**Grade** **^13^**	**%** **^12^**	**Grade** **^13^**
**Region 1**										
PL	-	-	-	-	9.0	1.2	-	-	4.0	1.0
Farm	2.6	3.1	2.8	2.2	50.5	1.5	1.3	2.5	45.9	2.7
Broodstock	4.2	3.4	3.8	1.8	63.7	1.4	1.4	1.0	39.0	1.4
Wild *P. vannamei*	-	-	20.5	3.0	70.5	1.8	-	-	133.3	1.2
Wild *P. stylirostris*	-	-	-	-	63.3	1.3	-	-	126.7	1.5
**Region 3**										
PL	-	-	-	-	14.4	1.0	-	-	-	-
Farm	-	-	-	-	**45.8**	**1.2**	-	-	58.3	1.1
Broodstock	-	-	-	-	**70.0**	**1.5**	-	-	-	-
Wild *P. monodon*	-	-	-	-	**60.0**	**1.0**	-	-	-	-
**Region 4**										
PL	-	-	-	-	**9.3**	**1.0**	-	-	-	-
Farm	-	-	29.3	1.8	**100**	**2.4**	-	-	-	-
Broodstock	-	-	-	-	**50.0**	**2.0**	-	-	-	-

^1^ The average hepatopancreas abnormalities (cell sloughing, hemocytic-melanized, and necrotic tubules) were recorded. ^2^ Atrophied/destroyed tubules, hemocytic enteritis. Intestinal epithelium with enteritis: observing the inflammation of intestinal walls with hemocyte infiltration. ^3^ Muscle necrosis, the presence of bacterial packs, and hemocyte infiltration. ^4^ Lymphoid organ spheroids or any pathology. ^5^ White spot syndrome virus lesions. ^6^ Hepanhamaparvovirus (DHPV) VIN. ^7^ Wenzhou shrimp virus 8 (WzSV8) VIN. ^8^ Infection by microsporidians in the hepatopancreas, antennal gland, and connective tissue of the stomach. ^9^ Sum of gregarines and nematodes in the intestine, hepatopancreas, etc. ^10^ Average weight (g). ^11^ Total number of animals analyzed. ^12^ Percentage of prevalence. ^13^ The average severity was adopted from [54] and simplified as follows: 0 = no lesions, 1 = lesions or infection present in <25% of area or organ or tissue sections, 2 = lesions or infection present in 25–50% of area or organ or tissue sections, 3 = lesions or infection present in 50–75% of area or organ or tissue sections, and 4 = lesions or infection present in >75% of area or organ or tissue sections [54]. ^14^ ND—not determined.

## Data Availability

The datasets generated and analyzed during the current study are available from the corresponding author upon reasonable request. Due to the sensitive nature of location-specific data, access may be restricted to ensure compliance with ethical, privacy, or commercial considerations.

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
