# Peer review of "Advanced Pathogen Monitoring in Penaeus vannamei from Three Latin American Regions: Passive Surveillance Part 2"

_viruses, 2025, doi:10.3390/v17020187_

Round 1
Reviewer 1 Report
Comments and Suggestions for Authors
In this paper, the author presented the second phase of a year-long investigation comparing 10 multiple PCR analyses and histological examinations to confirm the presence of characteristic lesions for each pathogen in three different regions of Latin America. As excellent as the first phase (DOI:10.1016/j.aqrep.2024.102092), this paper showed an amazing amount of work and meticulous work content. However, Minor Revision was decided before publication. The detailed comments are following:
1. As introduced in line 63, wild penaeid shrimps (P. vannamei, P. stylirostris and P. monodon) were sampled and all results contained these three species. It was suggested to change the “Penaeus vannamei” in title.
2. Some format errors were frequent, please revise.
Abbreviation error: P. vannamei in line 14; P. stylirostris, P. monodon in line 15; MHBV, DIV1, WSSV, IHHNV in line 35; PvSV, PvNV, CMNV, IMNV, YHV, TSV ,MrNV in line 36; AHPND, EHP in line 38; EVE in line 46 and so on. Almost complete spellings introduced in Section 2.3.
Italic error: Spiroplasma in line 490.
Font size error in line 511.
3. All tables should be three-line table. Table 8 showed different clarity in manuscript. And please check the align format.
4. Several figure legends need revision. Figure 1 legend “1 Nunan et al. (2003) 2 Potts et al. (2020) 3 Aranguren et al. (2010).” was not showed in figure. Line 354, 365, 370, 413 had wrong marked letters.
Author Response
Plase check the corrected manuscript all questions were already answered
REVIEWER 1
Line 2-3: The title 'Advanced Pathogen Monitoring...' is somewhat misleading. The techniques used in the MS for monitoring the pathogen are basic and involve Level I and III diagnostics combined. What did the authors mean by the word 'Advanced' in the title? Is it because of the use of end-point PCR (molecular diagnostic) or different primer level comparisons? It would be better if they could substantiate this; otherwise, it is better to change the title according to the results conveyed, considering the title of the first publication.
Comprehensive Pathogen Monitoring in Penaeus vannamei from Three Latin American Regions: Passive Surveillance Part 2
Line 36: What is the rationale for the inclusion of MrNV in the pathogen list to survey? Is there any report on MrNV detection in Penaeus species from the Latin American region? I could see only one reference (ref #50) reporting MrNV detection in P. vannamei in Asia. I request the authors to comment on these questions.
Macrobrachium rosenbergii nodavirus (MrNV), the causative agent of white tail disease in freshwater prawns, has been experimentally transmitted to three marine penaeid shrimp species: Penaeus indicus, P. japonicus, and P. monodon. Although these species did not exhibit mortality, MrNV was detectable via PCR. Notably, when larvae of Macrobrachium were fed tissue from these penaeids, mortality ensued. Given that MrNV is a nodavirus responsible for white tail disease, we aimed to ensure its absence in the region.
https://doi.org/10.1016/j.aquaculture.2006.02.053
Line 39-45: Considering the Part 1 surveillance results and publication, the statement given by the authors is acceptable. However, it is understood that the detection primers are designed based on the sequence in the conservative region of the genome. This means 80-90% of that sequence has a lower chance of acquiring mutations compared to other parts of the genome. Also, these regions are evolutionarily derived in many cases. Under these circumstances, how certain are the authors that geographical attributes, host, or other mentioned factors can reliably influence the mutation? Are the differences in detection between the regional samples using the same primer statistically significant? Is it possible to attribute this to the testing method, other laboratory reasons, or manual errors? The authors could give some comments on this
The reviewer may have a point; there could be laboratory or sampling errors. However, in Part One, which has already been published, each primer (for the same pathogen) was tested on an average of 115 samples, and in this second part, 140 samples were analyzed—resulting in a total of more than 250 samples showing the same pattern. Additionally, the goal was to evaluate what the industry is currently using and verify that it is being applied correctly.
An interesting study cited in both versions of the manuscript is by Safeena et al. (2012), titled "Molecular Biology and Epidemiology of Hepatopancreatic Parvovirus of Penaeid Shrimp" (Indian J Virol. 23:191-202), which demonstrates geographical differences in DHPV (a DNA virus). The most important DNA viruses in shrimp are WSSV, IHHNV, and DHPV, with
genome sizes of approximately 300, 4.1, and 6 Kb, respectively. On the other hand, the primary RNA viruses—TSV, WzSV8, and YHV—have genomes of approximately 9, 10, and 22 Kb. Of these, WSSV, with its large genome, is the most stable; its primer sequences have remained effective for almost 30 years. Another example of the industry's stubbornness is the continued use of a single primer to detect IHHNV (a DNA virus), despite sufficient information in the literature demonstrating that this virus is integrated into the shrimp genome as endogenous viral elements (EVEs). In the worst-case scenario, all four primers should be used (as employed in this version and in Part 1) to discriminate between actual viral presence and detection of EVEs.
The presence of IHHNV-derived EVEs in shrimp genomes can lead to false-positive results when using single-primer PCR methods, as these primers may amplify the EVE sequences instead of indicating an active infection. Using multiple primers targeting different regions of the IHHNV genome can help distinguish between EVEs and active viral infections, thereby improving diagnostic accuracy.
methodology.
Line 84-85: If the authors used different individual shrimps for the PCR analysis and histopathology, then how did they correlate the PCR results with histology? Are they considered on a population basis? Otherwise, how did they exclude the individual pathogen load and resultant pathology variations? The sentence is somewhat spurious. The conclusion drawn from the result might be spurious if the shrimp sample subjected to PCR was negative, but the shrimp subjected to histology showed some pathology. The authors are requested to rectify the sentence to provide more clarity.
Shrimp organs are very small, making it impractical to fix the entire animal to avoid post-mortem histological fixation issues while also collecting samples for PCR analysis. In both studies, it was noted that in larval stages, a minimum of 1 gram of sample is collected for histology and PCR, which equates to approximately 200 to 400 larvae per gram, depending on size. For juveniles and adults, more individuals are sampled to determine prevalence rates. The primary organ for pathological study in shrimp is the hepatopancreas. We fix the entire hepatopancreas for histology and PCR because some pathogens are unevenly distributed within the organ, and subsampling could lead to false negatives.
There has not been any sample where the histology is positive and the PCR is negative; in fact, the most important pathologies, such as wzSV8 and DHPV, go hand in hand. On the other hand, there are some cases of positive PCR but negative histologies.
Additionally, reporting pathogens based on geographic location can pose national challenges, as importing countries may use this information to impose trade barriers or lower prices. Many shrimp pathogens have originated in Asia, yet some Asian countries still enforce such barriers. Consequently, shrimp farmers often refrain from sharing disease outbreaks or pathological findings to avoid complications. As a small research laboratory, we analyze more than what is requested to assess the sector's overall epidemiological
status. The competent authority is responsible for official reporting. We cover all expenses, including travel, sample transportation, and reporting, at no cost. If we were involved in official reporting, we would likely lose access to farms, hatcheries, or sample submissions.
It's important to note that proper fixation techniques are crucial for accurate histological analysis. For instance, smaller shrimp (<1g) may be fixed directly by immersing in appropriate fixative for 24 hours, while larger shrimps require longer fixation times.
- Line 136-158: Did the authors get any reference for the tissue pooling for the mentioned pathogens? I wonder if they used the same number of animals (PL, adult, or broodstock – not mentioned clearly for the pooling) per pool for every pathogen. A viral particle dilution issue may arise. If one of the five individuals is actually positive but all others are negative, the pathogen may get diluted and fall below the detection level of the PCR (it's conventional here). The authors are advised to comment on this fact.
https://www.woah.org/fileadmin/Home/eng/Internationa_Standard_Setting/docs/pdf/2.2.02_IHHN.pdf
https://www.woah.org/fileadmin/Home/eng/Health_standards/aahm/current/chapitre_wsd.pdf
https://www.woah.org/fileadmin/Home/eng/Internationa_Standard_Setting/docs/pdf/2.2.04_TAURA.pdf
- Line 159-177: How did the authors cross-check or confirm the results of the conventional PCR they used? Did they use a real-time PCR method for that (many of the pathogens have real-time PCR as the gold standard)? Even in the case of negative results from the end-point PCR for a pathogen, one of the random pools of samples could be subjected to cross-check with the reported real-time PCR to ensure the reliability of the whole results in the MS. For example, in the case of TSV detection, there is an updated real-time PCR method from Moody et al., 2023 that could be used.
Normally, we have DNA or samples stored at -80°C in our bank that are positive, and we always run positive and negative samples as controls. Additionally, the sample we use is generally larger than the recommended size to ensure the maximum amount of tissue is available for extraction. Pathogens like Vibrio spp. or AHPND are easier to work with because we have collections of over 50 Vibrio strains, and primers are calibrated with these. For pathogens that are very common, such as WzSV8 and DHPV, paired samples are stored to compare PCR results with histology, and dilutions are determined using available material.
We always aim to use methods recommended by the WHOA (WSSV, IHHNV, IMNV, YHV, TSV, MRNV, DHPV, DIV1, AHPND, EHP, NHPB). For other pathogens, we test published methods and select the most sensitive ones.
- Line 206-209: In Table 2, one PL was detected positive for HPV out of three samples tested. Have you observed any histological changes concerning the HPV infection? If the PL subjected to PCR detection and histology are different, then how do the authors account for Query no. 3?
In Table 2, in Region 3, there is a sample of PL (33%) positive for DHPV using a specific set of primers (the most specific among all tested). However, in Table 8, the typical inclusion body of DHPV was not observed in larvae (unlike WzSV8, which was detected in PL by both PCR and histology). The larval samples come from the same tank pool, with at least 2 grams separated: 1 gram is fixed in Davidson’s fixative, and the other gram is preserved in alcohol (for PCR). Each gram is estimated to contain 200 to 400 animals, making it an excellent pool for both PCR and histology. In histology, up to 25 PL are generally observed per sample, with larval samples being much more comprehensive than those from larger animals.
Speculating further, it’s important to remember that DHPV (DNA) and WzSV8 (RNA) are both enteric viruses. In PL, the hepatopancreas is in extensive development (for instance, the relationship between the anterior diverticulum and the hepatopancreas changes with age), and the posterior diverticulum develops after PL10. It’s possible that these tissues are still too primitive for viral replication, although the viruses may be present in low numbers without causing enough inclusion bodies. This is often seen with WSSV, where many samples test positive by PCR without showing histological damage
- Figure 2: The prevalence derived from these results could be an average estimate of prevalence rather than a true prevalence. Over a one-year period, WSSV prevalence increased from 10% to ~30% (a 20% increase). AHPND prevalence also increased by more than 10%. In the case of EHP, it was also 20%. Were the authors able to identify any outbreak issues during the survey? Since the geographical locations are not revealed in the MS, it is difficult to believe the potential geographical variations in connection with the genotypic occurrence of those pathogens in different regions.
Completely agree. I’d like to remind you that the title of part one is Passive Surveillance for Shrimp Pathogens in Penaeus vannamei Submitted from Three Regions of Latin America.
I quote the rationale of the part 1 study
This study was not an epidemiological study but a simple prevalence analysis of various shrimp pathogens in randomly provided samples from hatcheries, farms, maturation units and wild animals from 3 different regions in Latin America from October 2022 to April 2023. We decided to test different primers because of our experience that histological findings sometimes did not match the PCR results, e.g., the lack of correspondence between histology and PCR for intracellular bacteria, microsporidia and DHPV. We also tested different sets of primers, such as those for the spore wall protein (SWP) gene of EHP, because of reports of cross reactions from
existing PCR detection methods that target the EHP small subunit ribosomal RNA gene (Jaroenlak et al. 2016, Tang et al. 2017, Dhar et al. 2023, Peña- Navarro et al. 2023). We performed tests with DNA from closely related microsporidia and other aquatic organisms from different regions in Latin America. We also tested different primers of the new virus WzSV8, as there are several reports of it from Asia to America. In this regard, Fredriksson et al. (2013), working with samples from a wastewater plant, found that the choice of PCR primers had an impact on assessments of bacterial community diversity and population dynamics. In addition, Klindworth et al. (2013) found that out of the 175 primers and 512 primer pairs checked, only 10 could be recommended as broad-range primers. In conclusion, even commonly used single primers exhibited significant differences in overall coverage and phylum spectrum.
- Line 462-463: The authors could outsource the sample for ISH if they preserved the sample accordingly. The resultant data could have provided more explanation for the digestive system alterations observed as discussed.
We currently have a project to test ISH, but with a different pathology. ISH has been used for some pathologies, but in this study, it would have been a herculean task to try to apply it to all pathogens and to develop each set, requiring a massive budget. That’s why we chose to use different staining techniques, such as DNA and RNA staining, to demonstrate the inclusion bodies of DHPV and WzSV8 in different organs.
- Line 508-513: Whether the authors observed EVE implications only in IHHNV samples, it could also be possible in the case of HPV and WSSV as reported. How does this account for?
It is very likely that there are other types of EVEs, just like in humans, as the shrimp genome must have thousands. For starters, the EVE of WSSV has already been documented. To be honest, what has been published by Mahidol University (Thailand) and NACAU (Saudi Arabia) are projects aiming not necessarily for a patent but rather for the development of an animal with an economic purpose. Not everything that has been published is the most accurate (that’s my personal feeling). I am not trying to criticize these authors—in fact, I know them and believe they are doing an extraordinary job—but it’s not an area they openly share because of the economic interests involved.
Reviewer 2 Report
Comments and Suggestions for Authors
Please see the attached file. Thank you!

Author Response
REVIEWER 1
Line 2-3: The title 'Advanced Pathogen Monitoring...' is somewhat misleading. The techniques used in the MS for monitoring the pathogen are basic and involve Level I and III diagnostics combined. What did the authors mean by the word 'Advanced' in the title? Is it because of the use of end-point PCR (molecular diagnostic) or different primer level comparisons? It would be better if they could substantiate this; otherwise, it is better to change the title according to the results conveyed, considering the title of the first publication.
Comprehensive Pathogen Monitoring in Penaeus vannamei from Three Latin American Regions: Passive Surveillance Part 2
Line 36: What is the rationale for the inclusion of MrNV in the pathogen list to survey? Is there any report on MrNV detection in Penaeus species from the Latin American region? I could see only one reference (ref #50) reporting MrNV detection in P. vannamei in Asia. I request the authors to comment on these questions.
Macrobrachium rosenbergii nodavirus (MrNV), the causative agent of white tail disease in freshwater prawns, has been experimentally transmitted to three marine penaeid shrimp species: Penaeus indicus, P. japonicus, and P. monodon. Although these species did not exhibit mortality, MrNV was detectable via PCR. Notably, when larvae of Macrobrachiumwere fed tissue from these penaeids, mortality ensued. Given that MrNV is a nodavirus responsible for white tail disease, we aimed to ensure its absence in the region.
https://doi.org/10.1016/j.aquaculture.2006.02.053
Line 39-45: Considering the Part 1 surveillance results and publication, the statement given by the authors is acceptable. However, it is understood that the detection primers are designed based on the sequence in the conservative region of the genome. This means 80-90% of that sequence has a lower chance of acquiring mutations compared to other parts of the genome. Also, these regions are evolutionarily derived in many cases. Under these circumstances, how certain are the authors that geographical attributes, host, or other mentioned factors can reliably influence the mutation? Are the differences in detection between the regional samples using the same primer statistically significant? Is it possible to attribute this to the testing method, other laboratory reasons, or manual errors? The authors could give some comments on this
The reviewer may have a point; there could be laboratory or sampling errors. However, in Part One, which has already been published, each primer (for the same pathogen) was tested on an average of 115 samples, and in this second part, 140 samples were analyzed—resulting in a total of more than 250 samples showing the same pattern. Additionally, the goal was to evaluate what the industry is currently using and verify that it is being applied correctly.
An interesting study cited in both versions of the manuscript is by Safeena et al. (2012), titled "Molecular Biology and Epidemiology of Hepatopancreatic Parvovirus of Penaeid Shrimp" (Indian J Virol. 23:191-202), which demonstrates geographical differences in DHPV (a DNA virus). The most important DNA viruses in shrimp are WSSV, IHHNV, and DHPV, with genome sizes of approximately 300, 4.1, and 6 Kb, respectively. On the other hand, the primary RNA viruses—TSV, WzSV8, and YHV—have genomes of approximately 9, 10, and 22 Kb. Of these, WSSV, with its large genome, is the most stable; its primer sequences have remained effective for almost 30 years. Another example of the industry's stubbornness is the continued use of a single primer to detect IHHNV (a DNA virus), despite sufficient information in the literature demonstrating that this virus is integrated into the shrimp genome as endogenous viral elements (EVEs). In the worst-case scenario, all four primers should be used (as employed in this version and in Part 1) to discriminate between actual viral presence and detection of EVEs.
The presence of IHHNV-derived EVEs in shrimp genomes can lead to false-positive results when using single-primer PCR methods, as these primers may amplify the EVE sequences instead of indicating an active infection. Using multiple primers targeting different regions of the IHHNV genome can help distinguish between EVEs and active viral infections, thereby improving diagnostic accuracy.
methodology.
Line 84-85: If the authors used different individual shrimps for the PCR analysis and histopathology, then how did they correlate the PCR results with histology? Are they considered on a population basis? Otherwise, how did they exclude the individual pathogen load and resultant pathology variations? The sentence is somewhat spurious. The conclusion drawn from the result might be spurious if the shrimp sample subjected to PCR was negative, but the shrimp subjected to histology showed some pathology. The authors are requested to rectify the sentence to provide more clarity.
Shrimp organs are very small, making it impractical to fix the entire animal to avoid post-mortem histological fixation issues while also collecting samples for PCR analysis. In both studies, it was noted that in larval stages, a minimum of 1 gram of sample is collected for histology and PCR, which equates to approximately 200 to 400 larvae per gram, depending on size. For juveniles and adults, more individuals are sampled to determine prevalence rates. The primary organ for pathological study in shrimp is the hepatopancreas. We fix the entire hepatopancreas for histology and PCR because some pathogens are unevenly distributed within the organ, and subsampling could lead to false negatives.
There has not been any sample where the histology is positive and the PCR is negative; in fact, the most important pathologies, such as wzSV8 and DHPV, go hand in hand. On the other hand, there are some cases of positive PCR but negative histologies.
Additionally, reporting pathogens based on geographic location can pose national challenges, as importing countries may use this information to impose trade barriers or lower prices. Many shrimp pathogens have originated in Asia, yet some Asian countries still enforce such barriers. Consequently, shrimp farmers often refrain from sharing disease outbreaks or pathological findings to avoid complications. As a small research laboratory, we analyze more than what is requested to assess the sector's overall epidemiological status. The competent authority is responsible for official reporting. We cover all expenses, including travel, sample transportation, and reporting, at no cost. If we were involved in official reporting, we would likely lose access to farms, hatcheries, or sample submissions.
It's important to note that proper fixation techniques are crucial for accurate histological analysis. For instance, smaller shrimp (<1g) may be fixed directly by immersing in appropriate fixative for 24 hours, while larger shrimps require longer fixation times.
- Line 136-158: Did the authors get any reference for the tissue pooling for the mentioned pathogens? I wonder if they used the same number of animals (PL, adult, or broodstock – not mentioned clearly for the pooling) per pool for every pathogen. A viral particle dilution issue may arise. If one of the five individuals is actually positive but all others are negative, the pathogen may get diluted and fall below the detection level of the PCR (it's conventional here). The authors are advised to comment on this fact.
https://www.woah.org/fileadmin/Home/eng/Internationa_Standard_Setting/docs/pdf/2.2.02_IHHN.pdf
https://www.woah.org/fileadmin/Home/eng/Health_standards/aahm/current/chapitre_wsd.pdf
https://www.woah.org/fileadmin/Home/eng/Internationa_Standard_Setting/docs/pdf/2.2.04_TAURA.pdf
- Line 159-177: How did the authors cross-check or confirm the results of the conventional PCR they used? Did they use a real-time PCR method for that (many of the pathogens have real-time PCR as the gold standard)? Even in the case of negative results from the end-point PCR for a pathogen, one of the random pools of samples could be subjected to cross-check with the reported real-time PCR to ensure the reliability of the whole results in the MS. For example, in the case of TSV detection, there is an updated real-time PCR method from Moody et al., 2023 that could be used.
Normally, we have DNA or samples stored at -80°C in our bank that are positive, and we always run positive and negative samples as controls. Additionally, the sample we use is generally larger than the recommended size to ensure the maximum amount of tissue is available for extraction. Pathogens like Vibrio spp. or AHPND are easier to work with because we have collections of over 50 Vibrio strains, and primers are calibrated with these. For pathogens that are very common, such as WzSV8 and DHPV, paired samples are stored to compare PCR results with histology, and dilutions are determined using available material.
We always aim to use methods recommended by the WHOA (WSSV, IHHNV, IMNV, YHV, TSV, MRNV, DHPV, DIV1, AHPND, EHP, NHPB). For other pathogens, we test published methods and select the most sensitive ones.
- Line 206-209: In Table 2, one PL was detected positive for HPV out of three samples tested. Have you observed any histological changes concerning the HPV infection? If the PL subjected to PCR detection and histology are different, then how do the authors account for Query no. 3?
In Table 2, in Region 3, there is a sample of PL (33%) positive for DHPV using a specific set of primers (the most specific among all tested). However, in Table 8, the typical inclusion body of DHPV was not observed in larvae (unlike WzSV8, which was detected in PL by both PCR and histology). The larval samples come from the same tank pool, with at least 2 grams separated: 1 gram is fixed in Davidson’s fixative, and the other gram is preserved in alcohol (for PCR). Each gram is estimated to contain 200 to 400 animals, making it an excellent pool for both PCR and histology. In histology, up to 25 PL are generally observed per sample, with larval samples being much more comprehensive than those from larger animals.
Speculating further, it’s important to remember that DHPV (DNA) and WzSV8 (RNA) are both enteric viruses. In PL, the hepatopancreas is in extensive development (for instance, the relationship between the anterior diverticulum and the hepatopancreas changes with age), and the posterior diverticulum develops after PL10. It’s possible that these tissues are still too primitive for viral replication, although the viruses may be present in low numbers without causing enough inclusion bodies. This is often seen with WSSV, where many samples test positive by PCR without showing histological damage
- Figure 2: The prevalence derived from these results could be an average estimate of prevalence rather than a true prevalence. Over a one-year period, WSSV prevalence increased from 10% to ~30% (a 20% increase). AHPND prevalence also increased by more than 10%. In the case of EHP, it was also 20%. Were the authors able to identify any outbreak issues during the survey? Since the geographical locations are not revealed in the MS, it is difficult to believe the potential geographical variations in connection with the genotypic occurrence of those pathogens in different regions.
Completely agree. I’d like to remind you that the title of part one is Passive Surveillance for Shrimp Pathogens in Penaeus vannamei Submitted from Three Regions of Latin America.
I quote the rationale of the part 1 study
This study was not an epidemiological study but a simple prevalence analysis of various shrimp pathogens in randomly provided samples from hatcheries, farms, maturation units and wild animals from 3 different regions in Latin America from October 2022 to April 2023. We decided to test different primers because of our experience that histological findings sometimes did not match the PCR results, e.g., the lack of correspondence between histology and PCR for intracellular bacteria, microsporidia and DHPV. We also tested different sets of primers, such as those for the spore wall protein (SWP) gene of EHP, because of reports of cross reactions from existing PCR detection methods that target the EHP small subunit ribosomal RNA gene (Jaroenlak et al. 2016, Tang et al. 2017, Dhar et al. 2023, Peña- Navarro et al. 2023). We performed tests with DNA from closely related microsporidia and other aquatic organisms from different regions in Latin America. We also tested different primers of the new virus WzSV8, as there are several reports of it from Asia to America. In this regard, Fredriksson et al. (2013), working with samples from a wastewater plant, found that the choice of PCR primers had an impact on assessments of bacterial community diversity and population dynamics. In addition, Klindworth et al. (2013) found that out of the 175 primers and 512 primer pairs checked, only 10 could be recommended as broad-range primers. In conclusion, even commonly used single primers exhibited significant differences in overall coverage and phylum spectrum.
- Line 462-463: The authors could outsource the sample for ISH if they preserved the sample accordingly. The resultant data could have provided more explanation for the digestive system alterations observed as discussed.
We currently have a project to test ISH, but with a different pathology. ISH has been used for some pathologies, but in this study, it would have been a herculean task to try to apply it to all pathogens and to develop each set, requiring a massive budget. That’s why we chose to use different staining techniques, such as DNA and RNA staining, to demonstrate the inclusion bodies of DHPV and WzSV8 in different organs.
- Line 508-513: Whether the authors observed EVE implications only in IHHNV samples, it could also be possible in the case of HPV and WSSV as reported. How does this account for?
It is very likely that there are other types of EVEs, just like in humans, as the shrimp genome must have thousands. For starters, the EVE of WSSV has already been documented. To be honest, what has been published by Mahidol University (Thailand) and NACAU (Saudi Arabia) are projects aiming not necessarily for a patent but rather for the development of an animal with an economic purpose. Not everything that has been published is the most accurate (that’s my personal feeling). I am not trying to criticize these authors—in fact, I know them and believe they are doing an extraordinary job—but it’s not an area they openly share because of the economic interests involved.

Reviewer 3 Report
Comments and Suggestions for Authors
This manuscript provides the advanced pathogen monitoring in penaeus vannamei from three latin american regions. WzSV8 was found to be widespread among shrimp in all regions, including both farm-raised and wild populations. Histo-pathological analysis indicated that shrimp typically presented coinfections, such as WzSV8, decapod hepanhamaparvovirus (DHPV), chronic midgut inflammation, and tu-bule distension/epithelial atrophy, consistent with the toxicity of Pir A/B or another bac-terial toxin. Bacterial muscle necrosis was also found in some regions. While the work is very interesting, the manuscript lacks important details needed to evaluate the validity of the study. Major concerns are detailed below.
1. Abbreviations should be defined or written in full when they first appear. Such as "DHPV" in line 35, please double check all the text to find similar errors and correct them.
2. in introduction, please briefly describe all pathogens.
3. Some key sample information is missing. In the materials and methods section, the author should add: how many farms are there in total, which ones are hatcheries, broodstock centers, farms or wild animals, how many samples were collected from each farm, their size, health status, etc.
4. Please provide all primers used for testing and their sources.
5. Academic article tables should use a three line table.
6. Figures 1 and 2 lack significance analysis.
7. The author should indicate how many samples were taken for histopathology and the ratio of positive signals.
8. Please unify the format of references in the article, including the author's name, the case of words in the title of the article, the writing of the name of the journal, and the page number.
Author Response
REVIEWER 2
This manuscript provides the advanced pathogen monitoring in penaeus vannamei from three latin american regions. WzSV8 was found to be widespread among shrimp in all regions, including both farm-raised and wild populations. Histo-pathological analysis indicated that shrimp typically presented coinfections, such as WzSV8, decapod hepanhamaparvovirus (DHPV), chronic midgut inflammation, and tu-bule distension/epithelial atrophy, consistent with the toxicity of Pir A/B or another bac-terial toxin. Bacterial muscle necrosis was also found in some regions. While the work is very interesting, the manuscript lacks important details needed to evaluate the validity of the study. Major concerns are detailed below.
- Abbreviations should be defined or written in full when they first appear. Such as "DHPV" in line 35, please double check all the text to find similar errors and correct them.
We included the most relevant keywords.
- in introduction, please briefly describe all pathogens. Done
- Some key sample information is missing. In the materials and methods section, the author should add: how many farms are there in total, which ones are hatcheries, broodstock centers, farms or wild animals, how many samples were collected from each farm, their size, health status, etc.
I did not include it because it occupies a full page, and readers can refer to Part 1, which has already been published. However, I am happy to include it if required
- Please provide all primers used for testing and their sources.
I did not include it because it occupies a full page, and readers can refer to Part 1, which has already been published. However, I am happy to include it if required.
- Academic article tables should use a three-line table.
I'm sorry, but I don't quite understand what you mean. Could you clarify?
- Figures 1 and 2 lack significance analysis. Please confirm if I need to delete them???
- The author should indicate how many samples were taken for histopathology and the ratio of positive signals.
I included this rationale in Part 1, but I can also include it here if required.
Rationale of the study
This study was not an epidemiological study but a simple prevalence analysis of various shrimp pathogens in randomly provided samples from hatcheries, farms, maturation units and wild animals from 3 different regions in Latin America from October 2022 to April 2023. We decided to test different primers because of our experience that histological findings sometimes did not match the PCR results, e.g., the lack of correspondence between histology and PCR for intracellular bacteria, microsporidia and DHPV. We also tested different sets of primers, such as those for the spore wall protein (SWP) gene of EHP, because of reports of cross reactions from existing PCR detection methods that target the EHP small subunit ribosomal RNA gene (Jaroenlak et al. 2016, Tang et al. 2017, Dhar et al. 2023, Peña- Navarro et al. 2023). We performedtests with DNA from closely related microsporidia and other aquatic organisms from different regions in Latin America. We also tested different primers of the new virus WzSV8, as there are several reports of it from Asia to America. In this regard, Fredriksson et al. (2013), working with samples from a wastewater plant, found that the choice of PCR primers had an impact on assessments of bacterial community diversity and population dynamics. In addition, Klindworth et al. (2013) found that out of the 175 primers and 512 primer pairs checked, only 10 could be recommended as broad-range primers. In conclusion, even commonly used single primers exhibited significant differences in overall coverage and phylum spectrum.
- Please unify the format of references in the article, including the author's name, the case of words in the title of the article, the writing of the name of the journal, and the page number.
I'm sorry, but I don't quite understand what you mean. Could you clarify?

Round 2
Reviewer 3 Report
Comments and Suggestions for Authors
The author did not respond to my questions.
Author Response
Plase check the new corrected manuscript besides all questions were already answered.
REVIEWER 2
This manuscript provides the advanced pathogen monitoring in penaeus vannamei from three latin american regions. WzSV8 was found to be widespread among shrimp in all regions, including both farm-raised and wild populations. Histo-pathological analysis indicated that shrimp typically presented coinfections, such as WzSV8, decapod hepanhamaparvovirus (DHPV), chronic midgut inflammation, and tu-bule distension/epithelial atrophy, consistent with the toxicity of Pir A/B or another bac-terial toxin. Bacterial muscle necrosis was also found in some regions. While the work is very interesting, the manuscript lacks important details needed to evaluate the validity of the study. Major concerns are detailed below.
- Abbreviations should be defined or written in full when they first appear. Such as "DHPV" in line 35, please double check all the text to find similar errors and correct them.
We included the most relevant keywords.
2. in introduction, please briefly describe all pathogens. Done
3. Some key sample information is missing. In the materials and methods section, the author should add: how many farms are there in total, which ones are hatcheries, broodstock centers, farms or wild animals, how many samples were collected from each farm, their size, health status, etc.
DONE
4. Please provide all primers used for testing and their sources.
DONE
5. Academic article tables should use a three-line table.
DONE
6. Figures 1 and 2 lack significance analysis. DELETED
7. The author should indicate how many samples were taken for histopathology and the ratio of positive signals.
DONE
Rationale of the study
This study was not an epidemiological study but a simple prevalence analysis of various shrimp pathogens in randomly provided samples from hatcheries, farms, maturation units and wild animals from 3 different regions in Latin America from October 2022 to April 2023. We decided to test different primers because of our experience that histological findings sometimes did not match the
PCR results, e.g., the lack of correspondence between histology and PCR for intracellular bacteria, microsporidia and DHPV. We also tested different sets of primers, such as those for the spore wall protein (SWP) gene of EHP, because of reports of cross reactions from existing PCR detection methods that target the EHP small subunit ribosomal RNA gene (Jaroenlak et al. 2016, Tang et al. 2017, Dhar et al. 2023, Peña- Navarro et al. 2023). We performed tests with DNA from closely related microsporidia and other aquatic organisms from different regions in Latin America. We also tested different primers of the new virus WzSV8, as there are several reports of it from Asia to America. In this regard, Fredriksson et al. (2013), working with samples from a wastewater plant, found that the choice of PCR primers had an impact on assessments of bacterial community diversity and population dynamics. In addition, Klindworth et al. (2013) found that out of the 175 primers and 512 primer pairs checked, only 10 could be recommended as broad-range primers. In conclusion, even commonly used single primers exhibited significant differences in overall coverage and phylum spectrum.
8. Please unify the format of references in the article, including the author's name, the case of words in the title of the article, the writing of the name of the journal, and the page number.
DONE
